# Parameter-free Statistically Consistent Interpolation: Dimension-independent Convergence Rates for Hilbert kernel regression

## Abstract

Previously, statistical textbook wisdom has held that interpolation of noisy training data will lead to poor generalization. However, recent work has shown that this is not true and that good generalization can be obtained with function fits that interpolate training data. This could explain why overparameterized deep nets with zero or small training error do not necessarily overfit and could generalize well. Data interpolation schemes have been exhibited that are provably Bayes optimal in the large sample limit and achieve the theoretical lower bounds for excess risk (Statistically Consistent Interpolation) in any dimension. These interpolation schemes are non-parametric Nadaraya-Watson style estimators with singular kernels, which exhibit statistical consistency in any data dimension for large sample sizes. The recently proposed weighted interpolating nearest neighbors scheme (wiNN) is in this class, as is the previously studied Hilbert kernel interpolation scheme. In the Hilbert scheme, the regression function estimator for a set of labelled data pairs, $(x_i, y_i) \in \mathbb{R}^d \times \mathbb{R}$, $i = 0, ..., n$, has the form $\hat{f}(x) = \sum_i y_i w_i(x)$, where $w_i(x) = \|x - x_i\|^{-d} / \sum_j \|x - x_j\|^{-d}$. This interpolating function estimator is unique in being entirely free of parameters and does not require bandwidth selection. While statistical consistency was previously proven for this scheme, the precise convergence rates for the finite sample risk were not established. Here, we carry out a comprehensive study of the asymptotic finite sample behavior of the Hilbert kernel regression scheme and prove a number of relevant theorems. We prove under broad conditions that the excess risk of the Hilbert regression estimator is asymptotically equivalent pointwise to $\sigma^2(x)/\ln(n)$ where $\sigma^2(x)$ is the noise variance. We also show that the excess risk of the plugin classifier is upper bounded by $2|f(x) - 1/2|^{1-\alpha} (1 + \varepsilon)^\alpha \sigma^\alpha(x)(\ln(n))^{-\frac{\alpha}{2}}$, for any $0 < \alpha < 1$, where $f$ is the regression function $x \mapsto \mathbb{E}[y|x]$. Our proofs proceed by deriving asymptotic equivalents of the moments of the weight functions $w_i(x)$ for large $n$, for instance for $\beta > 1$, $\mathbb{E}[w_i^\beta(x)] \sim_{n \to \infty} ((\beta - 1)n \ln(n))^{-1}$. We further derive an asymptotic equivalent for the Lagrange function and explicitly exhibit the nontrivial extrapolation properties of this estimator. Notably, the convergence rates are independent of data dimension and the excess risk is dominated by the noise variance. The bias term, for which we also give precise asymptotic estimates, is always subleading when the density of data at the considered point is strictly positive. If this local density is zero, we show that the bias term does not vanish in the limit of a large data set and we compute its limit explicitly. Finally, we present heuristic arguments for a universal $w^{-2}$ power-law behavior of the probability density of the weights in the large $n$ limit.

# 1  Introduction

Data interpolation and statistical regression of noisy data are both classical subjects but their domain of application have been disjoint until recently. Scattered data interpolation techniques [1] are generally used for clean data. On the other hand, when supervised learning or statistical regression techniques are applied to noisy data, in general smoothing or regularization methods are applied to prevent training data interpolation, as the latter is believed to lead to poor generalization [2]. However, accumulating empirical evidence from overparameterized deep networks has shown that data interpolation (equivalently, zero error on the training set) does not automatically imply poor generalization [3, 4]. This has in turn given rise to a rapidly growing body of theoretical work to understand how and why noisy data interpolation can still lead to good generalization [5, 6, 7, 8, 9, 10, 11, 12, 13, 14, 15].

A key observations in this regard is the phenomenon of Statistically Consistent Interpolation [16], i.e., regression function estimation that interpolates training data but also generalizes as well as possible by achieving the Bayes limit for expected generalization error (risk) when the sample size becomes large. This hints at a rich set of theoretical questions at the interface between the disciplines of scattered data interpolation and supervised learning, that have only begun to be addressed. In particular, there has been comparatively little study of the generalization error or risk of interpolating learners. Computation of generalization error bounds in machine learning often relies on the capacity of the class of fitting functions [17], however such model complexity based bounds are not tight enough to be useful for interpolating learners [4]. For nonparametric interpolation approaches such as that considered here, it is also not clear what model complexity means. Thus, there is a need for other approaches to understanding the generalization behavior of nonparametric interpolating learners, including more direct treatments of the generalization error for specific interpolation schemes so as to gain better theoretical understanding. The current paper addresses this need.

We present a detailed analysis of the finite-sample risk of an interpolating learner with intriguing theoretical properties, the Hilbert kernel estimator (Devroye *et. al.* [18]). A unique property of this Nadaraya-Watson (NW) style estimator [19, 20] is that it is fully parameter-free and does not have any bandwidth or scale parameter. It is global and uses all data points for each estimate: the associated kernel is a power law and thus scale-free. Although statistical consistency of this estimator was proven [18] when it was proposed, there has been no systematic analysis of the associated convergence rates and asymptotic finite sample behavior. We provide this analysis in the present study.

**Related work** The only other interpolation scheme we are aware of, that is proven to be statistically consistent in arbitrary dimensions under general conditions, is the recently proposed weighted interpolating nearest neighbors method (wiNN) [7], which is also a NW estimator utilizing a singular power law kernel of a very similar form but with two important differences: a finite number of neighbors $k$ is utilized (rather than all data points), and the power law exponent $\delta$ of the NW kernel satisfies $0 < \delta < d/2$ rather than $\delta = d$. To achieve consistency $k$ has to scale appropriately with sample size. Despite the superficial resemblance, the wiNN and Hilbert Kernel estimators have quite different convergence rates, as we will see from the results of this paper. Also worth mentioning is the Shepard interpolation scheme [21] originally proposed for interpolation of 2D geospatial data sets, also a NW style interpolating estimator, though used in the context of scattered data interpolation. In scattered data interpolation [1], the focus is generally on the approximation error (corresponding to the "bias" term in our analysis below). The approximation error of the Shepard scheme has been analyzed [22] but as we will see below the risk for Hilbert kernel interpolation is dominated by the noise or "variance" term. In contrast with wiNN or Hilbert kernel interpolation, other interpolating learning methods such as simplex interpolation [7] or ridgeless kernel regression [11] are generally not statistically consistent in fixed finite dimension [8].

**Summary of results of this paper** Notation and assumptions pertaining to this summary are defined in the problem setup section below. We prove under broad conditions that the excess risk of the Hilbert regression estimator is asymptotically equivalent pointwise to $\sigma^2(x)/\ln(n)$ where $\sigma^2(x)$ is the noise variance. We also show that the excess risk of the plugin classifier is upper bounded by $2|f(x) - 1/2|^{1-\alpha}(1+\varepsilon)^\alpha \sigma^\alpha(x)(\ln(n))^{-\frac{\alpha}{2}}$, for any $0 < \alpha < 1$, where $f$ is the regression function $x \mapsto \mathbb{E}[y|x]$. Our proofs proceed by deriving asymptotic equivalents of the moments of the weight functions $w_i(x)$ for large $n$, for instance for $\beta > 1$, $\mathbb{E}[w_i^\beta(x)] \sim_{n\to\infty} ((\beta - 1)n\ln(n))^{-1}$. We further derive an asymptotic equivalent for the Lagrange function and explicitly exhibit the nontrivial

extrapolation properties of this estimator. Notably, the convergence rates are independent of data dimension and the excess risk is dominated by the noise variance. The bias term, for which we also give precise asymptotic estimates, is always subleading when the density of data at the considered point is strictly positive. If this local density is zero, we show that the bias term does not vanish in the limit of a large data set and we compute its limit explicitly. Finally, we present heuristic arguments for a universal $w^{-2}$ power-law behavior of the probability density of the weights in the large $n$ limit.

# 2  Problem setup

**Notation, Definitions, Statistical Model** We model the labelled training data set $(x_0, y_0), \ldots, (x_n, y_n)$ as $n + 1$ *i.i.d.* observations of a random vector $(X, Y)$ with values in $\mathbb{R}^d \times \mathbb{R}$ for regression, and with values in $\mathbb{R}^d \times \{0, 1\}$ for binary classification. Due to the independence property, the collection $X_0, \ldots, X_n$ has the product density $\prod_{i=0}^{n} \rho(x_i)$. We will denote by $\mathbb{E}$ an expectation over the collection of $n + 1$ random vectors and by $\mathbb{E}_X$ the expectation over the collection $X_0, \ldots, X_n$. An expectation over the same collection while holding $X_i = x_i$ will be denoted $\mathbb{E}_{X|x_i}$. The regression function $f : \mathbb{R}^d \to \mathbb{R}$ is defined as the conditional mean of $Y$ given $X = x$, $f(x) := \mathbb{E}[Y \mid X = x]$ and the conditional variance function is $\sigma^2(x) := \mathbb{E}[|Y - f(X)|^2 | X = x]$. $f$ minimizes the expected value of the mean squared prediction error (risk under squared loss), $f = \arg\min \mathcal{R}_{\text{sq}}(h)$ where $\mathcal{R}_{\text{sq}}(h) := \mathbb{E}[(h(X) - Y)^2]$. Given any regression estimator $\hat{f}(x)$ the corresponding risk can be decomposed as $\mathbb{E}[\mathcal{R}_{\text{sq}}(\hat{f}(X))] = \mathcal{R}_{\text{sq}}(f) + \mathbb{E}[(\hat{f}(X) - f(X)^2]$. The excess risk is given by $\mathcal{R}_{\text{sq}}(\hat{f}) - \mathcal{R}_{\text{sq}}(f) = \mathbb{E}[(\hat{f}(X) - f(X))^2]$. For a consistent estimator this excess risk goes to zero as $n \to \infty$ and we are interested in characterizing the *rate* at which it goes to zero with increasing $n$ (note our sample size is $n + 1$ for notational simplicity but for large $n$ this does not change the rate).

In the case of binary classification, $Y \in \{0, 1\}$ and $f(x) = \mathbb{P}[Y = 1 \mid X = x]$. Let $F : \mathbb{R}^d \to \{0, 1\}$ denote the Bayes optimal classifier, defined by $F(x) := \theta(f(x) - 1/2)$ where $\theta(\cdot)$ is the Heaviside theta function. This classifier minimizes the risk $\mathcal{R}_{0/1}(h) := \mathbb{E}[\mathbb{1}_{\{h(X) \neq Y\}}] = \mathbb{P}(h(X) \neq Y)$ under zero-one loss. Given the regression estimator $\hat{f}$, we consider the plugin classifier $\hat{F}(x) = \theta(\hat{f}(x) - \frac{1}{2})$. The classification risk for the plugin classifier $\hat{F}$ is bounded as $\mathbb{E}[\mathcal{R}_{0/1}(\hat{F}(x))] - \mathcal{R}_{0/1}(F(x)) \leq 2\mathbb{E}[|\hat{f}(x) - f(x)|] \leq 2\sqrt{\mathbb{E}[(\hat{f}(x) - f(x))^2]}$.

Finally, we define two sequences $a_n, b_n > 0$, $n \in \mathbb{N}$, to be asymptotically equivalent for $n \to +\infty$, denoted $a_n \sim_{n \to +\infty} b_n$, if the limit of their ratio exists and $\lim_{n \to \infty} a_n/b_n = 1$.

In summary, our work will focus on the estimation of asymptotic equivalents for $\mathbb{E}[(\hat{f}(x) - f(x))^2]$ and other relevant quantities as this determines the rate at which the excess risk goes to zero for regression, and bounds the rate at which the excess risk goes to zero for classification.

**Assumptions.** We define the support $\Omega$ of the density $\rho$ as $\Omega = \{x \in \mathbb{R}^d / \rho(x) > 0\}$, the closed support $\bar{\Omega}$ as the closure of $\Omega$, and $\Omega^\circ$ as the interior of $\Omega$. Our results will not assume any compactness condition on $\Omega$ or $\bar{\Omega}$. The boundary of $\Omega$ is then defined as $\partial\Omega = \bar{\Omega} \setminus \Omega^\circ$. We assume that $\rho$ has a finite variance $\sigma_\rho^2$. In addition, we will most of the time assume that the density $\rho$ is continuous at the considered point $x \in \Omega^\circ$, and in some cases, $x \in \partial\Omega \cap \Omega$.

For the regression function $f$, we will obtain results assuming either of the following conditions

- $C_{\text{Cont}}^f$: $f$ is continuous at the considered $x$,

- $C_{\text{Holder}}^f$: for all $x \in \Omega^\circ$, there exist $\alpha_x > 0$, $K_x > 0$, and $\delta_x > 0$, such that
  $x' \in \Omega$ and $\|x - x'\| \leq \delta_x \implies |f(x) - f(x')| \leq K_x \|x - x'\|^{\alpha_x}$
  (local Hölder smoothness condition),

where condition $C_{\text{Holder}}^f$ is obviously stronger than $C_{\text{Cont}}^f$. In addition, we will always assume a growth condition for the regression function $f$:

- $C_{\text{Growth}}^f$: $\int \rho(y) \frac{f^2(y)}{1 + \|y\|^{2d}} \, d^d y < \infty$.

As for the variance function $\sigma$, we will obtain results assuming either that $\sigma$ is bounded or satisfies a growth condition similar to the one above

- $C_{\text{Bound}}^{\sigma}$: there exists $\sigma_0^2 \geq 0$, such that, for all $x \in \Omega$, we have $\sigma^2(x) \leq \sigma_0^2$,

- $C_{\text{Growth}}^{\sigma}$: $\int \rho(y) \frac{\sigma^2(y)}{1+\|y\|^{2d}} \, d^d y < \infty$.

When we will assume condition $C_{\text{Growth}}^{\sigma}$ (obviously satisfied when $\sigma^2$ is bounded), we will also assume a continuity condition $C_{\text{Cont}}^{\sigma}$ for $\sigma$ at the considered $x$.

Note that all our results can be readily extended in the case where $x \in \partial\Omega = \bar{\Omega} \setminus \Omega^{\circ}$ but keeping the condition $\rho(x) > 0$ (i.e., $x \in \partial\Omega \cap \Omega$), and assuming the continuity at $x$ of $\rho$ as seen as a function restricted to $\Omega$, i.e., $\lim_{y \in \Omega \to x} \rho(y) = \rho(x)$. Useful examples are when the support $\Omega$ of $\rho$ is a $d$-dimensional sphere or hypercube and $x$ is on the surface of $\Omega$ (but still with $\rho(x) > 0$). To guarantee these results for $x \in \partial\Omega \cap \Omega$, we need also to assume the continuity at $x$ of $f$, and assume that $\Omega$ is smooth enough near $x$, so that there exists a strictly positive local solid angle $\omega_x$ defined by

$$\omega_x = \lim_{r \to 0} \frac{1}{V_d \rho(x) r^d} \int_{\|x-y\| \leq r} \rho(y) \, d^d y = \lim_{r \to 0} \frac{1}{V_d r^d} \int_{y \in \Omega / \|x-y\| \leq r} d^d y, \tag{1}$$

where $V_d = S_d/d = \pi^{d/2}/\Gamma(d/2+1)$ is the volume of the unit ball in $d$ dimensions, and the second inequality results from the continuity of $\rho$ at $x$. If $x \in \Omega^{\circ}$, we have $\omega_x = 1$, while for $x \in \partial\Omega$, we have $0 \leq \omega_x \leq 1$. For instance, if $x$ is on the surface of a sphere or on the interior of a face of a hypercube (and in general, when the boundary near $x$ is locally an hyperplane), we have $\omega_x = \frac{1}{2}$. If $x$ is a corner of a hypercube, we have $\omega_x = \frac{1}{2^d}$. From our methods of proof presented in the appendix, it should be clear that all our results for $x \in \Omega^{\circ}$ perfectly generalize to any $x \in \partial\Omega \cap \Omega$ for which $\omega_x > 0$, by simply replacing $V_d$ whenever it appears in our different results by $\omega_x V_d$.

**Hilbert kernel interpolating estimator and Bias-Variance decomposition.** The Hilbert kernel regression estimator $\hat{f}(x)$ is a Nadaraya-Watson style estimator employing a singular kernel:

$$w_i(x) \;\; = \;\; \frac{\|x - x_i\|^{-d}}{\sum_{j=0}^{n} \|x - x_j\|^{-d}}, \tag{2}$$

$$\hat{f}(x) \;\; = \;\; \sum_{i=0}^{n} w_i(x) y_i. \tag{3}$$

The weights $w_i(x)$ are also called Lagrange functions in the interpolation literature and satisfy the interpolation property $w_i(x_j) = \delta_{ij}$, where $\delta_{ij} = 1$, if $i = j$, and 0 otherwise. At any given point $x$, they provide a partition of unity so that $\sum_{i=0}^{n} w_i(x) = 1$. The mean squared error between the Hilbert estimator and the true regression function has a bias-variance decomposition (using the *i.i.d* condition and the earlier definitions)

$$\hat{f}(x) - f(x) \;\; = \;\; \sum_{i=0}^{n} w_i(x)[f(x_i) - f(x)] + \sum_{i=0}^{n} w_i(x)[y_i - f(x_i)], \tag{4}$$

$$\mathbb{E}[(\hat{f}(x) - f(x))^2] \;\; = \;\; \mathcal{B}(x) + \mathcal{V}(x), \tag{5}$$

$$(Bias) \;\; \mathcal{B}(x) \;\; = \;\; \mathbb{E}_X\Big[\Big(\sum_{i=0}^{n} w_i(x)[f(x_i) - f(x)]\Big)^2\Big], \tag{6}$$

$$(Variance) \;\; \mathcal{V}(x) \;\; = \;\; \mathbb{E}\Big[\sum_{i=0}^{n} w_i^2(x)[y_i - f(x_i)]^2\Big] = \mathbb{E}_X\Big[\sum_{i=0}^{n} w_i^2(x)\sigma^2(x_i)\Big]. \tag{7}$$

The present work derives asymptotic behaviors and bounds for the regression and classification risk of the Hilbert estimator for large sample size $n$. These results are derived by analyzing the large $n$ behaviors of the bias and variance terms, which in turn depend on the behavior of the moments of the weights or the Lagrange functions $w_i(x)$. For all these quantities, asymptotically equivalent forms are derived. The proofs exploit a simple integral form of the weight function and details are provided in the appendix, while the body of the paper provides the results and associated discussions.

## 3  Results

### 3.1  The weights, variance and bias terms

#### 3.1.1  Moments of the weights: large $n$ behavior

In this section, we consider the moments and the distribution of the weights $w_i(x)$ at a given point $x$. The first moment is simple to compute. Since the weights sum to 1 and $X_i$ are *i.i.d*, it follows that $\mathbb{E}_{X|x_i}[w_i(x)]$ are all equal and thus $\mathbb{E}_{X|x_i}[w_i(x)] = (n+1)^{-1}$. The other moments are much less trivial to compute and we prove the following theorem in the appendix A.2:

**Theorem 3.1.** *For $x \in \Omega^\circ$ (so that $\rho(x) > 0$), we assume $\rho$ continuous at $x$. Then, the moments of the weight $w_0(x)$ satisfy the following properties:*

- *For $\beta > 1$:*

$$\mathbb{E}\left[w_0^\beta(x)\right] \underset{n \to +\infty}{\sim} \frac{1}{(\beta-1)n\ln(n)}. \tag{8}$$

- *For $0 < \beta < 1$: defining $\kappa_\beta(x) := \int \frac{\rho(x+y)}{||y||^{\beta d}} d^d y < \infty$, we have*

$$\mathbb{E}\left[w_0^\beta(x)\right] \underset{n \to +\infty}{\sim} \frac{\kappa_\beta(x)}{(V_d\rho(x)n\ln(n))^\beta}. \tag{9}$$

- *For $\beta < 0$: all moments for $\beta \leq -1$ are infinite, and the moments of order $-1 < \beta < 0$ satisfy*

$$\mathbb{E}\left[w_0^\beta(x)\right] \leq 1 + n\,\kappa_{|\beta|}(x)\kappa_\beta(x), \tag{10}$$

*so that a sufficient condition for its existence is $\kappa_\beta(x) = \int \rho(x+y)||y||^{|\beta|d} d^d y < \infty$.*

Heuristically, the behavior of these moments are consistent with the random variable $W = w_0(x)$ having a probability distribution satisfying a scaling relation $P(W) = \frac{1}{W_n}p\left(\frac{W}{W_n}\right)$, with the scaling function $p$ having the universal tail (i.e., independent of $x$ and $\rho$), $p(w) \underset{w \to +\infty}{\sim} w^{-2}$, and a scale $W_n$ expected to vanish with $n$, when $n \to +\infty$. With this assumption, we can determine the scale $W_n$ by imposing the exact condition $\mathbb{E}[W] = 1/(n+1) \sim 1/n$:

$$\mathbb{E}[W] = \frac{1}{W_n}\int_0^1 p\left(\frac{W}{W_n}\right) W\, dW = W_n \int_0^{\frac{1}{W_n}} p(w)w\, dw \tag{11}$$

$$\sim W_n \int_1^{\frac{1}{W_n}} \frac{dw}{w} \sim -W_n \ln(W_n) \sim \frac{1}{n}, \tag{12}$$

leading to $W_n \sim \frac{1}{n\ln(n)}$. Then, the moment of order $\beta > 1$ is given by

$$\mathbb{E}[W^\beta] = \frac{1}{W_n}\int_0^1 p\left(\frac{W}{W_n}\right)W^\beta\, dW \sim W_n \int_0^1 W^{\beta-2}\, dW \underset{n \to +\infty}{\sim} \frac{1}{(\beta-1)n\ln(n)}, \tag{13}$$

which indeed coincides with the first result of Theorem 3.1. Our heuristic argument also suggests that in the case $0 < \beta < 1$, we have

$$\mathbb{E}[W] = \frac{1}{W_n}\int_0^1 p\left(\frac{W}{W_n}\right)W^\beta\, dW \underset{n \to +\infty}{\sim} \frac{\int_0^{+\infty} p(w)w^\beta\, dw}{(n\ln(n))^\beta}, \tag{14}$$

where the last integral converges since $p(w) \underset{w \to +\infty}{\sim} w^{-2}$ and $\beta < 1$. This result is perfectly consistent with Eq. (9) in Theorem 3.1, and suggests that $\int_0^{+\infty} p(w)w^\beta\, dw = \frac{\kappa_\beta(x)}{(V_d\rho(x))^\beta}$. Interestingly, for $0 < \beta < 1$, and contrary to the case $\beta > 1$, we find that the large $n$ equivalent of the moment is not universal and depends explicitly on $x$ and the density $\rho$. As for moments of order $-1 < \beta < 0$, we conjecture that they are still given by Eq. (9) (and equivalently, by Eq. (14)) provided they exist, and that the sufficient condition for their existence $\kappa_\beta(x) < \infty$ is hence also necessary, since $\kappa_\beta(x)$ also appears in Eq. (9). The fact that moments for $\beta \leq -1$ do not exist strongly suggests that $p(0) > 0$.

In fact, Eq. (14)) also suggests that all moments for $-1 < \beta < 0$ exist if and only if $0 < p(0) < \infty$. In the Fig. 2 of the appendix, we present numerical simulations confirming our scaling ansatz, the fact that $p(w) \underset{w \to +\infty}{\sim} w^{-2}$, and the quantitative prediction for $W_n$.

It is shown in Devroye *et al.* [18] that the Hilbert kernel regression estimate does not converge almost surely (*a.s.*) by giving a specific example. Insight can be gained into this lack of almost sure convergence by considering the weight function $w_0(x)$, for a sequence of independent training sample sets of increasing size $n + 1$. Let the corresponding sequence of weights be denoted as $\omega_n \in [0, 1]$. From Theorem 3.1, it is clear that $\omega_n$ converges to zero in probability, since the following Chebyshev bound holds (analogous to the bound on the regression risk):

$$\mathbb{P}(\omega_n > \varepsilon) \leq \frac{1 + \delta}{\varepsilon^2 n \ln(n)}, \tag{15}$$

for arbitrary $\varepsilon > 0$ and $\delta > 0$, and for $n$ larger than some constant $N_{x,\delta}$. Alternatively, one can exploit the fact that $\mathbb{E}[\omega_n] = \frac{1}{n+1}$, leading to $\mathbb{P}(\omega_n > \varepsilon) \leq \frac{1}{\varepsilon n}$, which is less stringent than Eq. (15) as far as the $n$-dependence is concerned, but is more stringent for the $\varepsilon$-dependence of the bounds.

Let us show heuristically that $\omega_n$ does not converge *a.s.* to zero. Consider the infinite sequence of events $\mathcal{E}_n \equiv \{\omega_n > \varepsilon\}$, $n \in \mathbb{N}$, and the corresponding infinite sum $\sum_n \mathbb{P}(\mathcal{E}_n) = \sum_n \mathbb{P}(\omega_n > \varepsilon)$. Exploiting our previous heuristic argument for the scaling form of the distribution of weights, we obtain

$$\mathbb{P}(\omega_n > \varepsilon) = \int_\varepsilon^1 \frac{1}{W_n} p\left(\frac{W}{W_n}\right) dW \sim \int_{\varepsilon n \ln n}^{n \ln n} \frac{dw}{w^2} \sim \frac{1 - \varepsilon}{\varepsilon \, n \ln(n)}. \tag{16}$$

Since $\sum_{n=2}^N \frac{1}{n \ln(n)} \sim \ln(\ln(N))$ is a divergent series, a Borel-Cantelli argument suggests that an infinite number of the events $\mathcal{E}_n$ (i.e., $\omega_n > \varepsilon$) must occur, which implies that $\omega_n$ does not converge *a.s.* to 0. Note that the weights are equal to 1 at the data points due to the interpolation condition, so that large weights occasionally occur, causing the lack of *a.s.* convergence.

### 3.1.2 Lagrange function: scaling limit

The expected value of the Lagrange functions $w_i(x)$ have a simple form in the large $n$ limit. Due to the *i.i.d.* condition the indices $i$ are exchangeable and we set $i = 0$ for the computation of the expected Lagrange function $L_0(x) = \mathbb{E}_{X|x_0}[w_0(x)]$. Thus, one of the sample points (denoted $x_0$) is held fixed and the other ones are averaged over in computing the expected Lagrange function. For $x_0 \neq x$ kept fixed, we have $\lim_{n \to \infty} L_0(x) = 0$. However, we show in the appendix A.3 that $L_0(x)$ takes a very simple form when taking a specific scaling limit:

**Theorem 3.2.** *For $x \in \Omega^\circ$, we assume $\rho$ continuous at $x$. Then, in the limit (denoted by $\lim_Z$), $n \to +\infty$, $\|x - x_0\|^{-d} \to +\infty$ (i.e., $x_0 \to x$), and such that $z_x(n, x_0) = V_d \rho(x)\|x - x_0\|^d n \log(n) \to Z$, the Lagrange function $L_0(x) = \mathbb{E}_{X|x_0}[w_0(x)]$ converges to a proper limit,*

$$\lim_Z L_0(x) = \frac{1}{1 + Z}. \tag{17}$$

The proof of this theorem shows that the relative error between $L_0(x)$ and $\frac{1}{1+Z}$ for finite but large $n$ and large $\|x - x_0\|^{-d}$, such that $z_x(n, x_0)$ remains close to $Z$, is $O(1/\ln(n))$.

Exploiting Theorem 3.2, we can use a simple heuristic argument to estimate the tail of the distribution of the random variable $W = w_0(x)$. Indeed, approximating $L_0(x)$ for finite but large $n$ by its asymptotic form $\frac{1}{1+z_x(n,x_0)}$, with $z_x(n, x_0) = V_d \rho(x) n \log(n)\|x - x_0\|^d$, we obtain

$$\int_W^1 P(W') \, dW' \quad \sim \quad \int \rho(x_0) \, \theta\left(\frac{1}{1 + V_d \rho(x) n \log(n)\|x - x_0\|^d} - W\right) d^d x_0, \tag{18}$$

$$\sim \quad V_d \rho(x) \int_0^{+\infty} \theta\left(\frac{1}{1 + V_d \rho(x) n \log(n) \, u} - W\right) du, \tag{19}$$

$$\sim \quad \frac{1}{n \ln(n) W} \quad \implies \quad P(W) \sim \frac{1}{n \ln(n) W^2}, \tag{20}$$

where $\theta(.)$ is the Heaviside function. This heuristic result is again perfectly consistent with our guess of the previous section that $P(W) = \frac{1}{W_n} p\left(\frac{W}{W_n}\right)$, with the scaling function $p$ having the universal

tail, $p(w) \underset{w \to +\infty}{\sim} w^{-2}$, and a scale $W_n \sim \frac{1}{n \ln(n)}$. Indeed, in this case and in the limit $n \to +\infty$, we

obtain that $P(W) \sim \frac{1}{W_n} \left( \frac{W_n}{W} \right)^2 \sim \frac{W_n}{W^2} \sim \frac{1}{n \ln(n) W^2}$, which is identical to the result of Eq. (20).

### 3.1.3 The variance term

A simple application of the result of Theorem 3.1 for $\beta = 2$ (see appendix A.4) allows us to bound
the variance term $\mathcal{V}(x) = \mathbb{E}\left[ \sum_{i=0}^{n} w_i^2(x)[y_i - f(x_i)]^2 \right]$ for a bounded variance function $\sigma^2$:

**Theorem 3.3.** *For $x \in \Omega^\circ$, $\rho$ continuous at $x$, $\sigma^2 \leq \sigma_0^2$, and for any $\varepsilon > 0$, there exists a constant $N_{x,\varepsilon}$ such that for $n \geq N_{x,\varepsilon}$, we have*

$$\mathcal{V}(x) \leq (1 + \varepsilon) \frac{\sigma_0^2}{\ln(n)}. \tag{21}$$

Relaxing the boundedness condition for $\sigma$, but assuming the continuity of $\sigma^2$ at $x$ along with a growth
condition, allows us to obtain a precise asymptotic equivalent of $\mathcal{V}(x)$, when $n \to +\infty$:

**Theorem 3.4.** *For $x \in \Omega^\circ$, $\sigma(x) > 0$, $\rho\sigma^2$ continuous at $x$, and assuming the condition $C_{\text{Growth}}^\sigma$, i.e., $\int \rho(y) \frac{\sigma^2(y)}{1 + \|y\|^{2d}} \, d^d y < \infty$, we have*

$$\mathcal{V}(x) \underset{n \to +\infty}{\sim} \frac{\sigma^2(x)}{\ln(n)}. \tag{22}$$

Note that if the mean variance $\int \rho(y)\sigma^2(y) \, d^d y < \infty$, which is in particular the case when $\sigma^2$ is
bounded over $\Omega$, then the condition $C_{\text{Growth}}^\sigma$ is in fact automatically satisfied.

### 3.1.4 The bias term

In appendix A.5, we prove the following three theorems for the bias term.

**Theorem 3.5.** *For $x \in \Omega^\circ$ (so that $\rho(x) > 0$), we assume that $\rho$ is continuous at $x$, and the conditions*

- *$C_{\text{Growth}}^f$: $\int \rho(y) \frac{f^2(y)}{1 + \|y\|^{2d}} \, d^d y < \infty$,*

- *$C_{\text{Holder}}^f$: there exist $\alpha_x > 0$, $K_x > 0$, and $\delta_x > 0$, such that*
  *$x' \in \Omega$ and $\|x - x'\| \leq \delta_x \implies |f(x) - f(x')| \leq K_x \|x - x'\|^{\alpha_x}$*
  *(local Hölder condition for $f$).*

*Moreover, we define $\kappa(x) = \int \rho(x + y) \frac{f(x+y) - f(x)}{\|y\|^d} \, d^d y$, where we have $|\kappa(x)| < \infty$.*

*Then, for $\kappa(x) \neq 0$, the bias term $\mathcal{B}(x) = \mathbb{E}_X \left[ \left( \sum_{i=0}^{n} w_i(x)[f(x_i) - f(x)] \right)^2 \right]$ satisfies*

$$\mathcal{B}(x) \underset{n \to +\infty}{\sim} \left( \mathbb{E}\left[ \hat{f}(x) \right] - f(x) \right)^2, \qquad \text{with} \quad \mathbb{E}\left[ \hat{f}(x) \right] - f(x) \underset{n \to +\infty}{\sim} \frac{\kappa(x)}{V_d \rho(x) \ln(n)}. \tag{23}$$

*In the non generic case $\kappa(x) = 0$, we have the weaker result*

$$\mathcal{B}(x) = \begin{cases} O\left( n^{-\frac{2\alpha_x}{d}} (\ln(n))^{-1 - \frac{2\alpha_x}{d}} \right), & \text{for } d > 2\alpha_x \\ O\left( n^{-1}(\ln(n))^{-1} \right), & \text{for } d = 2\alpha_x \\ O\left( n^{-1}(\ln(n))^{-2} \right), & \text{for } d < 2\alpha_x \end{cases} \tag{24}$$

Note that $\kappa(x) = 0$ is non generic but can still happen, even if $f$ is not constant. For instance, if $\Omega$ is a
sphere centered at $x$ or $\Omega = \mathbb{R}^d$, if $\rho(x+y) = \hat{\rho}(\|y\|)$ is isotropic around $x$, and if $f_x : y \mapsto f(x+y)$
is an odd function of $y$, then we indeed have $\kappa(x) = 0$ at this symmetric point $x$.

Interestingly, for $\kappa(x) \neq 0$, Eq. (23) shows that the bias $\mathcal{B}(x)$ is asymptotically dominated by the square of $\mathbb{E}\left[\hat{f}(x)\right] - f(x)$, showing that the fluctuations of $\mathbb{E}\left[\hat{f}(x)\right] - \sum_{i=0}^{n} w_i(x) f(x_i)$ are negligible compared to $\mathbb{E}\left[\hat{f}(x)\right] - f(x)$, in the limit $n \to +\infty$ and for $\kappa(x) \neq 0$.

One can relax the local Hölder condition, but at the price of a weaker estimate for $\mathcal{B}(x)$ which will however be enough to obtain strong results for the regression and classification risks (see below):

**Theorem 3.6.** *For $x \in \Omega^\circ$, we assume $\rho$ and $f$ continuous at $x$, and the growth condition $C_{\text{Growth}}^f$: $\int \rho(y) \frac{f^2(y)}{1+\|y\|^{2d}} d^d y < \infty$. Then, the bias term satisfies*

$$\mathcal{B}(x) = o\left(\frac{1}{\ln(n)}\right), \tag{25}$$

*or equivalently, for any $\varepsilon > 0$, there exists $N_{x,\varepsilon}$, such that for $n \geq N_{x,\varepsilon}$*

$$\mathcal{B}(x) \leq \frac{\varepsilon}{\ln(n)}. \tag{26}$$

Let us now consider a point $x \in \partial\Omega$ for which we have $\rho(x) = 0$ (note that $x \in \partial\Omega$ does not necessarily imply $\rho(x) = 0$). In appendix A.5, we show the following theorem for the expectation value of the estimator $\hat{f}(x)$ in the limit $n \to +\infty$:

**Theorem 3.7.** *For $x \in \partial\Omega$ such that $\rho(x) = 0$, we assume that $f$ and $\rho$ satisfy the conditions*

- *$C_{\text{Growth}}^f$: $\int \rho(y) \frac{|f(y)|}{1+\|y\|^d} d^d y < \infty$,*

- *$C_{\text{Holder}}^\rho$: there exist $\alpha_x > 0$, $K_x > 0$, and $\delta_x > 0$, such that*
  *$x' \in \Omega$ and $\|x - x'\| \leq \delta_x \implies |\rho(x')| \leq K_x \|x - x'\|^{\alpha_x}$*
  *(local Hölder condition for $\rho$).*

*Moreover, we define $\kappa(x) = \int \rho(x + y) \frac{f(x+y) - f(x)}{\|y\|^d} d^d y$ ($|\kappa(x)| < \infty$ under condition $C_{\text{Growth}}^f$), and $\lambda(x) = \int \frac{\rho(x+y)}{\|y\|^d} d^d y$ ($0 < \lambda(x) < \infty$ under condition $C_{\text{Holder}}^\sigma$). Then,*

$$\lim_{n \to +\infty} \mathbb{E}[\hat{f}(x)] - f(x) = \frac{\kappa(x)}{\lambda(x)}. \tag{27}$$

Hence, in the generic case $\kappa(x) \neq 0$ (see Theorem 3.5 and the discussion below it) and under condition $C_{\text{Holder}}^\rho$, we find that the bias does not vanish when $\rho(x) = 0$, and that the estimator $\hat{f}(x)$ does not converge to $f(x)$. When $\rho(x) = 0$, the scarcity of data near the point $x$ indeed prevents the estimator to converge to the actual value of $f(x)$. In appendix A.5, we show an example of a density $\rho$ continuous at $x$ and such that $\rho(x) = 0$, but not satisfying the condition $C_{\text{Holder}}^\rho$, and for which $\lim_{n \to +\infty} \mathbb{E}[\hat{f}(x)] = f(x)$, even if $\kappa(x) \neq 0$.

## 3.2 Asymptotic equivalent for the regression risk

In appendix A.6, we prove the following theorem establishing the asymptotic rate at which the excess risk goes to zero with large sample size $n$ for Hilbert kernel regression, under mild conditions that do not require $f$ or $\sigma$ to be bounded, but only to satisfy some growth conditions:

**Theorem 3.8.** *For $x \in \Omega^\circ$, we assume $\sigma(x) > 0$, $\rho$, $\sigma$, and $f$ continuous at $x$, and the growth conditions $C_{\text{Growth}}^\sigma$: $\int \rho(y) \frac{\sigma^2(y)}{1+\|y\|^{2d}} d^d y < \infty$ and $C_{\text{Growth}}^f$: $\int \rho(y) \frac{f^2(y)}{1+\|y\|^{2d}} d^d y < \infty$.*

*Then the following statements are true:*

- *The excess regression risk at the point $x$ satisfies*

$$\mathbb{E}[(\hat{f}(x) - f(x))^2] \underset{n \to +\infty}{\sim} \frac{\sigma^2(x)}{\ln(n)}. \tag{28}$$

- *The Hilbert kernel estimate converges pointwise to the regression function in probability. More specifically, for any $\delta > 0$, there exists a constant $N_{x,\delta}$, such that for any $\varepsilon > 0$, we have the following Chebyshev bound, valid for $n \geq N_{x,\delta}$,*

$$\mathbb{P}[|\hat{f}(x) - f(x)| \geq \varepsilon] \leq \frac{1+\delta}{\varepsilon^2} \frac{\sigma^2(x)}{\ln(n)}. \tag{29}$$

This theorem is a consequence of the corresponding asymptotically equivalent forms of the variance and bias terms presented above. Note that as long as $\rho(x) > 0$, the variance term dominates over the bias term and the regression risk has the same form as the variance term.

### 3.3 Rates for the plugin classifier

In appendix A.7, we prove the following theorem establishing the asymptotic rate at which the classification risk goes to zero with large sample size $n$ for Hilbert kernel regression:

**Theorem 3.9.** *For $x \in \Omega^\circ$, we assume $\sigma(x) > 0$, $\rho$, $\sigma$, and $f$ continuous at $x$. Then, the classification risk $\mathbb{E}[\mathcal{R}_{0/1}(\hat{F}(x))] - \mathcal{R}_{0/1}(F(x))$ vanishes for $n \to +\infty$.*

*More precisely, for any $\varepsilon > 0$, there exists $N_{x,\varepsilon}$, such that for any $n \geq N_{x,\varepsilon}$,*

$$0 \leq \mathbb{E}[\mathcal{R}_{0/1}(\hat{F}(x))] - \mathcal{R}_{0/1}(F(x)) \leq 2(1+\varepsilon)\frac{\sigma(x)}{\sqrt{\ln(n)}}, \tag{30}$$

*In addition, for any $0 < \alpha < 1$, the general inequality*

$$\mathbb{E}[\mathcal{R}_{0/1}(\hat{F}(x))] - \mathcal{R}_{0/1}(F(x)) \leq 2|f(x) - 1/2|^{1-\alpha} \, \mathbb{E}\left[|\hat{f}(x) - f(x)|^2\right]^{\frac{\alpha}{2}}, \tag{31}$$

*holds unconditionally and, for $n \geq N_{x,\varepsilon}$, leads to*

$$0 \leq \mathbb{E}[\mathcal{R}_{0/1}(\hat{F}(x))] - \mathcal{R}_{0/1}(F(x)) \leq 2|f(x) - 1/2|^{1-\alpha} \, (1+\varepsilon)^\alpha \frac{\sigma^\alpha(x)}{(\ln(n))^{\frac{\alpha}{2}}}. \tag{32}$$

For $0 < \alpha < 1$, Eq. (32) is weaker than Eq. (30) in terms of its dependence on $n$, but explicitly shows that the classification risk vanishes for $f(x) = 1/2$. This theorem does not require any growth condition for $f$ or $\sigma$, since both functions takes values in $[0,1]$ in the classification context.

### 3.4 Extrapolation behavior outside the support of $\rho$

We now take the point $x$ outside the closed support $\bar{\Omega}$ of the distribution $\rho$ (which excludes the case $\Omega = \mathbb{R}^d$). We are interested in the behavior of $\mathbb{E}\left[\hat{f}(x)\right]$ as $n \to +\infty$. In appendix A.8 we prove:

**Theorem 3.10.** *For $x \notin \bar{\Omega}$, we assume the growth condition $\int \rho(y)\frac{|f(y)|}{1+\|y\|^d} \, d^dy < \infty$. Then,*

$$\hat{f}_\infty(x) := \lim_{n \to +\infty} \mathbb{E}\left[\hat{f}(x)\right] = \frac{\int \rho(y)f(y)\|x - y\|^{-d} \, d^dy}{\int \rho(y)\|x - y\|^{-d} \, d^dy}, \tag{33}$$

*and $\hat{f}_\infty$ is continuous at all $x \notin \bar{\Omega}$.*

*In addition, if $\int \rho(y)|f(y)| \, d^dy < \infty$, and defining $d(x, \Omega) > 0$ as the distance between $x$ and $\Omega$, we have*

$$\lim_{d(x,\Omega) \to +\infty} \hat{f}_\infty(x) = \int \rho(y)f(y) \, d^dy. \tag{34}$$

*Finally, we consider $x_0 \in \partial\Omega$ such that $\rho(x_0) > 0$ (i.e., $x_0 \in \partial\Omega \cap \Omega$), and assume that $f$ and $\rho$ seen as functions restricted to $\Omega$ are continuous at $x_0$, i.e. $\lim_{y \in \Omega \to x_0} \rho(y) = \rho(x_0)$ and $\lim_{y \in \Omega \to x_0} f(y) = f(x_0)$. We also assume that the local solid angle $\omega_0 = \lim_{r \to 0} \frac{1}{V_d\rho(x_0)r^d} \int_{\|x_0 - y\| \leq r} \rho(y) \, d^dy$ exists and satisfies $\omega_0 > 0$. Then,*

$$\lim_{x \notin \bar{\Omega} \to x_0} \hat{f}_\infty(x) = f(x_0). \tag{35}$$

Eq. (34) shows that far away from $\Omega$ (which is possible to realize, for instance, when $\Omega$ is bounded), $\hat{f}_\infty(x)$ goes smoothly to the $\rho$-mean of $f$. Moreover, Eq. (35) establishes a continuity property for the extrapolation $\hat{f}_\infty$ at $x_0 \in \partial\Omega \cap \Omega$ under the stated conditions (remember that for $x \in \Omega^\circ$, we have $\lim_{n \to +\infty} \mathbb{E}\left[\hat{f}(x)\right] = f(x)$; see Theorem 3.5, and in particular Eq. (23)).

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
