# A  Proofs

## A.1  Preliminaries

In the following, $x \in \Omega^\circ$ so that $\rho(x) > 0$, and we will assume for simplicity that the distribution $\rho$ is continuous at $x$.

For the proof of our results, we will often exploit the following integral relation, valid for $\beta > 0$,

$$\frac{1}{\Gamma(\beta)} \int_0^{+\infty} t^{\beta-1} \mathrm{e}^{-t\,z}\, dt = z^{-\beta}. \tag{36}$$

In addition, we define

$$\psi(x,t) := \int \rho(x+y) \mathrm{e}^{-\frac{t}{||y||^d}}\, d^d y, \tag{37}$$

which will play a central role. We note that $\psi(x,0) = 1$, and that $t \mapsto \psi(x,t)$ is a continuous and strictly decreasing function of $t$. It is even infinitely differentiable at any $t > 0$, but not necessarily at $t = 0$. In fact, for a fixed $x$, controlling the behavior of $1 - \psi(x,t)$ when $t \to 0$ will be essential to obtain our results.

We show in Fig. 1 an example of the Hilbert kernel regression estimator in one dimension. Both the bias and the variance of the estimator can be visually seen, as well as the extrapolation behavior outside the data domain. Note that in higher dimensions, the sharp peaks would have rounded tops.

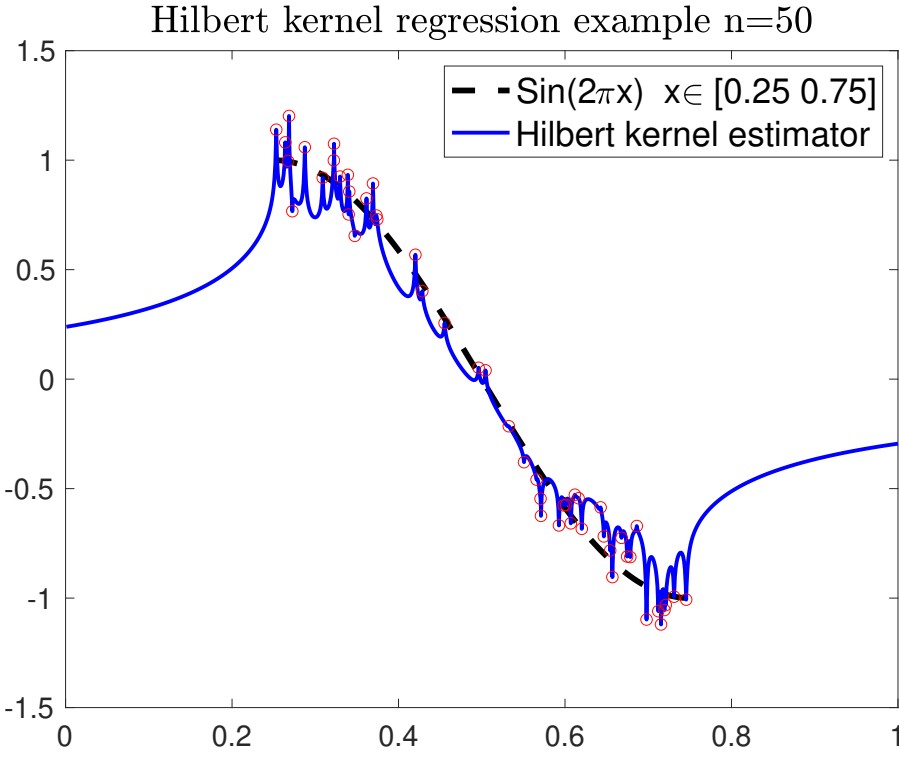

Figure 1: An example is shown of the Hilbert kernel regression estimator in one dimension, both within and outside the input data domain. A total of 50 samples $x_i$ were chosen uniformly distributed in the interval $[0.25 \quad 0.75]$ and $y_i = \sin(2\pi x_i) + n_i$ with the noise $n_i$ chosen *i.i.d.* Gaussian distributed $\sim N(0, 0.1)$. The sample points are circled, and the function $\sin(2\pi x)$ is shown with a dashed line within the data domain. The solid line is the Hilbert kernel regression estimator. Note the interpolation behavior within the data domain and the extrapolation behavior outside the data domain.

## A.2 Moments of the weights: large $n$ behavior

In this section, we provide a complete proof of Theorem 3.1. Several other theorems will use the very same method of proof and some basic steps will not be repeated in their proof.

Using Eq. (36) for $\beta > 0$, we can express powers of the weight function as

$$w_0^\beta(x) = \frac{1}{||x-x_0||^{\beta d}} \frac{1}{\Gamma(\beta)} \int_0^{+\infty} t^{\beta-1} e^{-t\,||x-x_0||^{-d} - t \sum_{i=1}^n ||x-x_i||^{-d}} \, dt. \tag{38}$$

By taking the expected value over the $n+1$ independent random variables $X_i$, we obtain

$$\mathbb{E}\left[w_0^\beta(x)\right] = \frac{1}{\Gamma(\beta)} \int_0^{+\infty} t^{\beta-1} \psi^n(x,t) \phi_\beta(x,t) \, dt, \tag{39}$$

with

$$\phi_\beta(x,t) := \int \rho(x+y) \frac{e^{-\frac{t}{||y||^d}}}{||y||^{\beta d}} \, d^d y, \tag{40}$$

which is also a strictly decreasing function of $t$, continuous at any $t > 0$ (in fact, infinitely differentiable for $t > 0$).

Note that the exchange of the integral over $t$ and over $\vec{x} = (x_0, x_1, ..., x_n)$ used to obtain Eq. (39) is justified by the Fubini theorem, by first noting that the function $\vec{x} \mapsto w_0^\beta(x) \prod_{i=0}^n \rho(x_i)$ is in $L^1(\mathbb{R}^d)$, since $0 \leq w_0^\beta(x) \leq 1$, and since $\rho$ is obviously in $L^1(\mathbb{R}^d)$. Moreover, the function $t \mapsto t^{\beta-1} \psi^n(x,t) \phi_\beta(x,t) > 0$ is also in $L^1(\mathbb{R})$. Indeed, we will show below that it decays fast enough when $t \to +\infty$ (see Eqs. (42-50)), ensuring the convergence of its integral at $+\infty$, and that it is bounded (and continuous) near $t = 0$ (see Eqs. (63-68)), ensuring that this function is integrable at $t = 0$.

For $\beta = 1$, $\phi_1 = -\partial_t \psi$, and we obtain $\mathbb{E}\left[w_0(x)\right] = \frac{1}{n+1}$, as expected. In the following, we first focus on the case $\beta > 1$, before addressing the cases $0 < \beta < 1$ and $\beta < 0$ at the very end of this section.

We now introduce $t_1$ and $t_2$ (to be further constrained later) such that $0 < t_1 < t_2$. We then express the integral of Eq. (39) as the sum of corresponding integrals $I_1 + I_{12} + I_2$. $I_1$ is the integral between $0$ and $t_1$, $I_{12}$ the integral between $t_1$ and $t_2$, and $I_2$ the integral between $t_2$ and $+\infty$. Thus, we have

$$I_1 \leq \mathbb{E}\left[w_0^\beta(x)\right] \leq I_1 + I_{12} + I_2, \tag{41}$$

provided these integral exists, which we will show below, by providing upper bounds for $I_2$ and $I_{12}$, and tight lower and upper bound for the leading term $I_1$.

*Bound for $I_2$*

For any $R \geq 1$, we can write the integral defining $\psi(x,t)$

$$\psi(x,t) = \int_{||y|| \leq R} + \int_{||y|| \geq R} \tag{42}$$

$$\leq e^{-\frac{t}{R^d}} + \int_{||y|| \geq R} \rho(x+y) \frac{||y||^2}{R^2} \, d^d y, \tag{43}$$

$$\leq e^{-\frac{t}{R^d}} + \frac{C_x}{R^2}, \tag{44}$$

with $C_x = \sigma_\rho^2 + ||x - \mu_\rho||^2$ depending on the mean $\mu_\rho$ and variance $\sigma_\rho^2$ of the distribution $\rho$. Similarly, for $\phi_\beta(x,t)$, we obtain the bound

$$\phi_\beta(x,t) \leq \frac{1}{R^{\beta d}} e^{-\frac{t}{R^d}} + \frac{C_x}{R^{2+\beta d}}, \tag{45}$$

valid for $t \geq \max(1, \beta)$ and $R \leq r_t$, where $r_t = (t/\beta)^{1/d} \geq 1$ is the location of the maximum of the function $r \mapsto \frac{e^{-\frac{t}{r^d}}}{r^{\beta d}}$.

We now set $R = t^{\frac{s}{d}}$, with $0 < s < 1$, and take $T_2' \geq \max(1, \beta, \beta^{1/(1-s)})$ (so that $1 \leq R \leq r_t$) is large enough such that the following conditions are satisfied for $t \geq t_2 \geq T_2'$,

$$\mathrm{e}^{-\frac{t}{R^d}} = \mathrm{e}^{-t^{1-s}} \quad \leq \quad \frac{C_x}{t^{\frac{2s}{d}}}, \tag{46}$$

$$\frac{1}{R^{\beta d}}\mathrm{e}^{-\frac{t}{R^d}} = \frac{1}{t^{\beta s}}\mathrm{e}^{-t^{1-s}} \quad \leq \quad \frac{C_x}{t^{\frac{2s}{d}+2\beta s}}. \tag{47}$$

Hence, for $t \geq t_2 \geq T_2'$, we obtain

$$\psi(x,t) \quad \leq \quad \frac{2C_x}{t^{\frac{2s}{d}}}, \tag{48}$$

$$\phi_\beta(x,t) \quad \leq \quad \frac{2C_x}{t^{\frac{2s}{d}+2\beta s}}. \tag{49}$$

In addition, we also impose $t_2 \geq T_2'' = (4C_x)^{d/(2s)}$, so that $\frac{2C_x}{t^{\frac{2s}{d}}} \leq \frac{1}{2}$, for any $t \geq T_2 = \max(T_2', T_2'')$. Finally, exploiting the resulting bounds for $\psi(x,t)$ and $\phi_\beta(x,t)$ for $s = 1/2$, we obtain the convergence of $I_2$ (which, along with the bounds for $I_1$ and $I_{12}$ below, justifies our use of Fubini theorem to obtain Eq. (39)) and the exact bound

$$I_2 = \frac{1}{\Gamma(\beta)} \int_{t_2}^{+\infty} t^{\beta-1}\psi^n(x,t)\phi_\beta(x,t)\, dt \leq \frac{d}{\Gamma(\beta)} \times \frac{1}{2^{n+1}(n+1)}, \tag{50}$$

for any given $t_2 \geq T_2$.

*Bound for $I_{12}$*

Again, exploiting the fact that $\psi(x,t)$ and $\phi_\beta(x,t)$ are strictly decreasing functions of $t$, we obtain

$$I_{12} \leq \frac{\phi_\beta(x,t_1)t_2^\beta}{\Gamma(\beta)} \times \psi^n(x,t_1), \tag{51}$$

where we note that $\psi(x,t_1) < 1$, for any $t_1 > 0$.

*Bound for $I_1$*

We first want to obtain bounds for $1 - \psi(x,t)$, where $0 \leq t \leq t_1$, with $t_1 > 0$ to be constrained below. In addition, exploiting the continuity of $\rho$ at $x$ and the fact that $\rho(x) > 0$, we introduce $\varepsilon$ satisfying $0 < \varepsilon < 1/4$, and define $\lambda > 0$ small enough so that the ball $B(x,\delta) \subset \Omega^\circ$, and $||y|| \leq \lambda \implies |\rho(x+y) - \rho(x)| \leq \varepsilon\rho(x)$. Exploiting this definition, we obtain the following lower and upper bounds

$$1 - \psi(x,t) \quad \geq \quad (1-\varepsilon)\rho(x)\int_{||y||\leq\lambda} \left(1 - \mathrm{e}^{-\frac{t}{||y||^d}}\right) d^d y, \tag{52}$$

$$1 - \psi(x,t) \quad \leq \quad (1+\varepsilon)\rho(x)\int_{||y||\leq\lambda} \left(1 - \mathrm{e}^{-\frac{t}{||y||^d}}\right) d^d y \tag{53}$$

$$+ \int_{||y||\geq\lambda} \rho(x+y)\left(1 - \mathrm{e}^{-\frac{t}{\lambda^d}}\right) d^d y, \tag{54}$$

$$\leq \quad (1+\varepsilon)\rho(x)\int_{||y||\leq\lambda} \left(1 - \mathrm{e}^{-\frac{t}{||y||^d}}\right) d^d y + \frac{t}{\lambda^d}. \tag{55}$$

The integral appearing in these bounds can be simplified by using radial coordinates:

$$\int_{||y||\leq\lambda} \left(1 - \mathrm{e}^{-\frac{t}{||y||^d}}\right) d^d y \quad = \quad S_d \int_0^\lambda \left(1 - \mathrm{e}^{-\frac{t}{r^d}}\right) r^{d-1}\, dr, \tag{56}$$

$$= \quad V_d t \int_{\frac{t}{\lambda^d}}^{+\infty} \frac{1 - \mathrm{e}^{-u}}{u^2}\, du, \tag{57}$$

where $S_d$ and $V_d = \frac{S_d}{d}$ are respectively the surface and the volume of the $d$-dimensional unit sphere and we have used the change of variable $u = \frac{t}{r^d}$.

We note that for $0 < z \leq 1$, we have

$$\int_z^{+\infty} \frac{1 - \mathrm{e}^{-u}}{u^2}\, du = -\ln(z) + \int_z^1 \frac{1 - u - \mathrm{e}^{-u}}{u^2}\, du + \int_1^{+\infty} \frac{1 - \mathrm{e}^{-u}}{u^2}\, du. \tag{58}$$

Exploiting this result and now imposing $t_1 \leq \lambda^d$, we have, for any $t \leq t_1$

$$\ln\left(\frac{C_-}{t}\right) \leq \int_{\frac{t}{\lambda^d}}^{+\infty} \frac{1 - \mathrm{e}^{-u}}{u^2}\, du \leq \ln\left(\frac{C_+}{t}\right), \tag{59}$$

$$\ln(C_-) = d\ln(\lambda) + \int_1^{+\infty} \frac{1 - \mathrm{e}^{-u}}{u^2}\, du, \tag{60}$$

$$\ln(C_+) = \ln(C_-) + \int_0^1 \frac{1 - u - \mathrm{e}^{-u}}{u^2}\, du. \tag{61}$$

Combining these bounds with Eq. (52) and Eq. (55), we have shown the existence of two $x$-dependent constants $D_\pm$ such that, for $0 \leq t \leq t_1 \leq \lambda^d$, we have

$$(1 - \varepsilon)V_d\rho(x)\, t\ln\left(\frac{D_-}{t}\right) \leq 1 - \psi(x,t) \leq (1 + \varepsilon)V_d\rho(x)\, t\ln\left(\frac{D_+}{t}\right). \tag{62}$$

In addition, we will also chose $t_1 < D_\pm/3$, such that the two functions $t\ln\left(\frac{D_\pm}{t}\right)$ are positive and strictly increasing for $0 \leq t \leq t_1$. $t_1$ is also taken small enough such that the two bounds in Eq. (62) are always less than 1/2, for $0 \leq t \leq t_1$ (both bounds vanish when $t \to 0$).

We now obtain efficient bounds for $\phi_\beta(x,t)$, for $0 \leq t \leq t_1$. Proceeding in a similar manner as above, we obtain

$$\phi_\beta(x,t) \geq (1 - \varepsilon)\rho(x)\int_{||y|| \leq \lambda} \frac{\mathrm{e}^{-\frac{t}{||y||^d}}}{||y||^{\beta d}}\, d^d y, \tag{63}$$

$$\phi_\beta(x,t) \leq (1 + \varepsilon)\rho(x)\int_{||y|| \leq \lambda} \frac{\mathrm{e}^{-\frac{t}{||y||^d}}}{||y||^{\beta d}}\, d^d y + \frac{1}{\lambda^{\beta d}}. \tag{64}$$

Again, the integral appearing in these bounds can be rewritten as

$$\int_{||y|| \leq \lambda} \frac{\mathrm{e}^{-\frac{t}{||y||^d}}}{||y||^{\beta d}}\, d^d y = S_d \int_0^\lambda r^{d(1-\beta)-1}\mathrm{e}^{-\frac{t}{r^d}}\, dr. \tag{65}$$

For $0 < \beta < 1$, the integral of Eq. (65) is finite for $t = 0$, ensuring the existence of $\phi_\beta(x,0)$ and the fact that $t \mapsto t^{\beta-1}\psi(x,t)\phi_\beta(x,t)$ belongs to $L^1(\mathbb{R})$ (hence, justifying our use of Fubini theorem for $0 < \beta < 1$). For $\beta > 1$, we have

$$\int_{||y|| \leq \lambda} \frac{\mathrm{e}^{-\frac{t}{||y||^d}}}{||y||^{\beta d}} = V_d\, t^{1-\beta}\int_{\frac{t}{\lambda^d}}^{+\infty} u^{\beta-2}\mathrm{e}^{-u}\, du. \tag{66}$$

$$\sim_{t \to 0} V_d\Gamma(\beta - 1)t^{1-\beta}. \tag{67}$$

This integral diverges when $t \to 0$ and the constant term $\lambda^{-\beta d}$ in Eq. (64) can be made as small as necessary (by a factor less than $\varepsilon$) compared to this leading integral term, for a small enough $t_1$. Similarly, we can choose $t_1$ small enough so that the integral Eq. (65) is approached by the asymptotic result of Eq. (67) up to a factor $\varepsilon$. Thus, we find that for $0 \leq t \leq t_1$, one has

$$(1 - 2\varepsilon)V_d\rho(x)\Gamma(\beta - 1)t^{1-\beta} \leq \phi_\beta(x,t) \leq (1 + 3\varepsilon)V_d\rho(x)\Gamma(\beta - 1)t^{1-\beta}. \tag{68}$$

This shows that $t^{\beta-1}\phi_\beta(x,t)$ has a smooth limit when $t \to 0$ so that, combined with the finite upper bound for $I_2$, $t \mapsto t^{\beta-1}\psi(x,t)\phi_\beta(x,t)$ belongs to $L^1(\mathbb{R})$, for $\beta > 1$, and hence for all $\beta > 0$. Hence, the use of the Fubini theorem to derive Eq. (39) has been justified.

Now combining the bounds for $\psi(x,t)$ and $\phi_\beta(x,t)$, we obtain

$$I_1 \geq (1 - 2\varepsilon)\frac{1}{\beta - 1}V_d\rho(x)\int_0^{t_1}\left(1 - (1 + \varepsilon)V_d\rho(x)\, t\ln\left(\frac{D_+}{t}\right)\right)^n dt, \tag{69}$$

$$I_1 \leq (1 + 3\varepsilon)\frac{1}{\beta - 1}V_d\rho(x)\int_0^{t_1}\left(1 - (1 - \varepsilon)V_d\rho(x)\, t\ln\left(\frac{D_-}{t}\right)\right)^n dt. \tag{70}$$

*Asymptotic behavior of $I_1$ and $\mathbb{E}\left[w_0^\beta(x)\right]$*

We will show below that

$$\int_0^{t_1} \left(1 - E_\pm t \ln\left(\frac{D_\pm}{t}\right)\right)^n dt \underset{n\to+\infty}{\sim} \frac{1}{E_\pm n \ln(n)}, \tag{71}$$

where $E_\pm = (1 \mp \varepsilon) V_d \rho(x)$. For a given $x$, and for $t_1$ and $t_2$ satisfying the requirements mentioned above, the upper bounds for $I_{12}$ (see Eq. (51)) and $I_2$ (see Eq. (50)) appearing in Eq. (41) both decay exponentially with $n$ and can hence be made arbitrarily small compared to $I_1$ which decays as $1/(n \ln(n))$.

Finally, assuming for now the result of Eq. (71) (to be proven below), we have obtained the exact asymptotic result

$$\mathbb{E}\left[w_0^\beta(x)\right] \underset{n\to+\infty}{\sim} \frac{1}{(\beta - 1)n \ln(n)}. \tag{72}$$

*Proof of Eq. (71)*

We are then left to prove the result of Eq. (71). First, we will use the fact that, for $0 \le z \le z_1 < 1$, one has

$$e^{-\mu z} \le 1 - z \le e^{-z}, \tag{73}$$

where $\mu = -\ln(1 - z_1)/z_1$. We can apply this result to the integral of Eq. (71), using $z_1^\pm = E_\pm t_1 \ln(D_\pm/t_1) > 0$. Note that $0 < t_1 < D_\pm/3$ and hence $z_1^\pm > 0$ can be made as close to 0 as desired, and the corresponding $\mu_\pm > 1$ can be made as close to 1 as desired. Thus, in order to prove Eq. (71), we need to prove the following equivalent

$$I_n = \int_0^{t_1} e^{-nEt \ln\left(\frac{D}{t}\right)} dt \underset{n\to+\infty}{\sim} \frac{1}{En \ln(n)}, \tag{74}$$

for an integral of the form appearing in Eq. (74). Let us mention again that $t_1$ has been taken small enough, so that the function $t \mapsto t \ln\left(\frac{D}{t}\right)$ is positive and strictly increasing (with its maximum at $t_{\max} = D/e < t_1$), for $0 \le t \le t_1$.

We now take $n$ large enough so that $\frac{\ln(n)}{n} < t_1$ and $E \ln(n) > 1$. One can then write

$$I_n = \frac{1}{n} \int_0^{\ln(n)} e^{-Eu \ln\left(\frac{Dn}{u}\right)} du + \int_{\frac{\ln(n)}{n}}^{t_1} e^{-nEt \ln\left(\frac{D}{t}\right)} dt = J_n + K_n, \tag{75}$$

$$J_n \le \frac{1}{n} \int_0^{1/E} e^{-Eu \ln(DEn)} du + \frac{1}{n} \int_{1/E}^{\ln(n)} e^{-Eu \ln\left(\frac{Dn}{\ln(n)}\right)} du, \tag{76}$$

$$\le \frac{1}{E n \ln(D En)} + \frac{\ln(n)}{D E n^2 \ln\left(\frac{Dn}{\ln(n)}\right)}, \tag{77}$$

$$K_n \le \int_{\frac{\ln(n)}{n}}^{+\infty} e^{-nEt \ln\left(\frac{D}{t_1}\right)} dt \le \frac{1}{E n^{1+E \ln\left(\frac{D}{t_1}\right)} \ln\left(\frac{D}{t_1}\right)}. \tag{78}$$

When $n \to +\infty$, we hence find that the upper bound $I_n^+$ of $I_n$ satisfies

$$I_n^+ \underset{n\to+\infty}{\sim} \frac{1}{E n \ln(DEn)} \underset{n\to+\infty}{\sim} \frac{1}{E n \ln(n)}. \tag{79}$$

Let us now prove a similar result for a lower bound of $I_n$ by considering $n$ large enough so that $nEt_1 > 1$, and by introducing $\delta$ satisfying $0 \le \delta < 1/e$:

$$I_n = \frac{1}{nE} \int_0^{nEt_1} e^{-u \ln(DEn) + u \ln(u)} du, \tag{80}$$

$$\ge \frac{1}{nE} \int_0^\delta e^{-u \ln(DEn) + \delta \ln(\delta)} du, \tag{81}$$

$$\ge \frac{e^{\delta \ln(\delta)}}{nE \ln(DEn)} \left(1 - (DEn)^{-\delta}\right) = I_n^-(\delta). \tag{82}$$

Hence, for any $0 \leq \delta < 1/e$ which can be made arbitrarily small, and for $n$ large enough, we find that $I_n \geq I_n^-(\delta)$, with

$$I_n^-(\delta) \underset{}{\sim} \frac{\mathrm{e}^{\delta \ln(\delta)}}{E\,n \ln(DEn)} \sim \frac{\mathrm{e}^{\delta \ln(\delta)}}{E\,n \ln(n)}. \tag{83}$$

Eq. (83) combined with the corresponding result of Eq. (79) for the upper bound $I_n^+$ finally proves Eq. (74), and ultimately, Eq. (72) and Theorem 3.1 for the asymptotic behavior of the moment $\mathbb{E}\left[w_0^\beta(x)\right]$, for $\beta > 1$.

*Moments of order $0 < \beta < 1$*

The integral representation Eq. (36) allows us to also explore moments of order $0 < \beta < 1$. In that case $\kappa_\beta(x) = \phi_\beta(x, 0) < \infty$ is finite, with

$$\kappa_\beta(x) = \int \frac{\rho(x + y)}{||y||^{\beta d}}\, d^d y. \tag{84}$$

By retracing the different steps of our proof in the case $\beta > 1$, it is straightforward to show that

$$\mathbb{E}\left[w_0^\beta(x)\right] \underset{n \to +\infty}{\sim} \frac{\kappa_\beta(x)}{\Gamma(\beta)} \int_0^{t_1} t^{\beta - 1} \mathrm{e}^{-nV_d \rho(x) t \ln\left(\frac{D_\pm}{t}\right)}\, dt, \tag{85}$$

$$\underset{n \to +\infty}{\sim} \frac{\kappa_\beta(x)}{(V_d \rho(x) n \ln(n))^\beta}, \tag{86}$$

where the equivalent for the integral can be obtained by exploiting the very same method used in our proof of Eq. (71) above, hence proving the second part of Theorem 3.1.

We observe that contrary to the universal result of Eq. (72) for $\beta$, the asymptotic equivalent for the moment of order $0 < \beta < 1$ is non universal and explicitly depends on $x$ and the distribution $\rho$.

*Moments of order $\beta < 0$*

Finally, moments of order $\beta < 0$ are unfortunately inaccessible to our methods relying on the integral relation Eq. (36), which imposes $\beta > 0$. We can however obtain a few rigorous results for these moments (see also the heuristic discussion just after Theorem 3.1).

Indeed, for $\beta = -1$, we have

$$\frac{1}{w_0(x)} = 1 + ||x - x_0||^d \sum_{i=1}^n \frac{1}{||x - x_i||^d}. \tag{87}$$

But since we have assumed that $\rho(x) > 0$, $\mathbb{E}[||x - x_i||^{-d}] = \int \frac{\rho(x+y)}{||y||^d}\, d^d y$ is infinite and moments of order $\beta < -1$ are definitely not defined.

As for the moment of order $-1 < \beta < 0$, it can be easily bounded,

$$\mathbb{E}\left[w_0^\beta(x)\right] \leq 1 + n \int \rho(x + y)||y||^{|\beta| d}\, d^d y \int \frac{\rho(x + y)}{||y||^{|\beta| d}}\, d^d y, \tag{88}$$

and a sufficient condition for its existence is $\kappa_\beta(x) = \int \rho(x + y)||y||^{|\beta| d}\, d^d y < \infty$ (the other integral, equal to $\kappa_{|\beta|}(x)$, is always finite for $|\beta| < 1$), which proves the last part of Theorem 3.1.

*Numerical distribution of the weights*

In the main text below Theorem 3.1, we presented an heuristic argument showing that the results of Theorem 3.1 and Theorem 3.2 (for the Lagrange function; that we prove below) were fully consistent with the weight $W = w_0(x)$ having a long-tailed scaling distribution,

$$P_n(W) = \frac{1}{W_n} p\left(\frac{W}{W_n}\right). \tag{89}$$

The scaling function $p$ was shown to have a universal tail $p(w) \sim w^{-2}$ and the scale $W_n$ was shown to obey the equation $-W_n \ln(W_n) = n^{-1}$. To the leading order for large $n$, we have

$W_n \sim \frac{1}{n \ln(n)}$, and we can solve this equation recursively to find the next order approximation, $W_n \sim \frac{1}{n \ln(n \ln(n))}$. In Fig .2, we present numerical simulations for the scaling distribution $p$ of the variable $w = W/W_n$, for $n = 65536$, using the estimate $W_n \approx \frac{1}{n \ln(n \ln(n))}$. We observe that $p(w)$ is very well approximated by the function $\hat{p}(w) = \frac{1}{(1+w)^2}$, confirming our non rigorous results. The

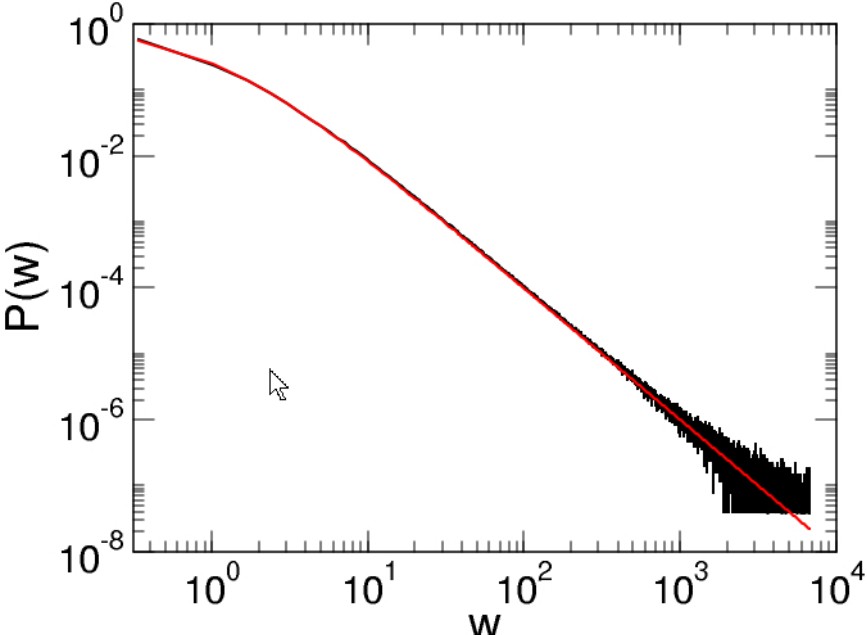

Figure 2: We plot the results of numerical simulations for the distribution $p$ of the scaling variable $w = \frac{W}{W_n}$, with $W_n \approx \frac{1}{n \ln(n \ln(n))}$, and for $n = 65536$ (black line). This is compared to $\hat{p}(w) = \frac{1}{(1+w)^2}$ (red line), which has the predicted universal tail $p(w) \sim w^{-2}$ for large $w$.

data were generated by drawing random values of $r_i^d = ||x - x_i||^d$ using $(n + 1)$ *i.i.d.* random variables $a_i$ uniformly distributed in $[0, 1[$, with the relation $r_i = [a_i/(1 - a_i)]^{1/d}$, and by computing the resulting weight $W = r_i^{-d} / \sum_{j=0}^n r_j^{-d}$. This corresponds to a distribution of $||x - x_i||$ given by $\rho(x - x_i) = 1/V_d/(1 + ||x - x_i||^d)^2$.

## A.3   Lagrange function: scaling limit

In this section, we prove Theorem 3.2 for the scaling limit of the Lagrange function $L_0(x) = \mathbb{E}_{X|x_0}[w_0(x)]$. Exploiting again Eq. (36), the expected Lagrange function can be written as

$$L_0(x) = ||x - x_0||^{-d} \int_0^{+\infty} \psi^n(x, t) e^{-t||x - x_0||^{-d}} \, dt, \tag{90}$$

where $\psi(x, t)$ is again given by Eq. (37).

For a given $t_1 > 0$, and remembering that $\psi(x, t)$ is a strictly decreasing function of $t$, with $\psi(x, 0) = 1$, we obtain

$$L_1 \leq L_0(x) \leq L_1 + L_2, \tag{91}$$

with

$$L_1 = ||x - x_0||^{-d} \int_0^{t_1} \psi^n(x, t) e^{-t||x - x_0||^{-d}} \, dt, \tag{92}$$

$$L_2 = e^{-t_1||x - x_0||^{-d}}. \tag{93}$$

For $\varepsilon > 0$ and a sufficiently small $t_1 > 0$ (see section A.2), we can use the bound for $\psi(x, t)$ obtained in section A.2, to obtain

$$L_1 \geq (1 - 2\varepsilon) \frac{1}{\|x - x_0\|^d} \int_0^{t_1} \left(1 - (1 + \varepsilon) V_d \rho(x) \, t \ln\left(\frac{D_+}{t}\right)\right)^n e^{-\frac{t}{\|x - x_0\|^d}} \, dt, \quad (94)$$

$$L_1 \leq (1 + 3\varepsilon) \frac{1}{\|x - x_0\|^d} \int_0^{t_1} \left(1 - (1 - \varepsilon) V_d \rho(x) \, t \ln\left(\frac{D_-}{t}\right)\right)^n e^{-\frac{t}{\|x - x_0\|^d}} \, dt. \quad (95)$$

Then, proceeding exactly as in section A.2, it is straightforward to show that $L_1$ can be bounded (up to factors $1 + O(\varepsilon)$) by the two integrals $L_1^{\pm}$

$$L_1^{\pm} = \frac{1}{\|x - x_0\|^d} \int_0^{t_1} e^{-n \, V_d \rho(x) \, t \ln\left(\frac{D_\pm}{t}\right) - \frac{t}{\|x - x_0\|^d}} \, dt. \quad (96)$$

Like in section A.2, we impose $t_1 < D_{\pm}/3$, such that the two functions $t \ln\left(\frac{D_\pm}{t}\right)$ are positive and strictly increasing for $0 \leq t \leq t_1$.

We now introduce the scaling variable $z(n, x_0) = V_d \rho(x) \|x - x_0\|^d n \log(n)$, so that

$$L_1^{\pm} = \frac{1}{\|x - x_0\|^d} \int_0^{t_1} e^{-\frac{t}{\|x - x_0\|^d}\left(1 + z \frac{\ln\left(D_\pm/t\right)}{\ln(n)}\right)} \, dt = \int_0^{\frac{t_1}{\|x - x_0\|^d}} e^{-u\left(1 + z \frac{\ln\left(D_\pm \|x - x_0\|^{-d}/u\right)}{\ln(n)}\right)} \, du, \quad (97)$$

where we have used the shorthand notation $z = z(n, x_0)$.

For a given real $Z \geq 0$, we now want to study the limit of $L_0(x)$ when $n \to \infty$, $\|x - x_0\|^{-d} \to +\infty$ (i.e., $x_0 \to x$), and such that $z(n, x_0) \to Z$, which we will simply denote $\lim_Z L_0(x)$. We note that $\lim_Z L_2 = 0$ (see Eq. (91) and Eq. (93)), so that we are left to show that $\lim_Z L_1^{\pm} = \frac{1}{1+Z} = \lim_Z L_0(x)$, which will prove Theorem 3.2.

Exploiting the fact that $u \ln(u) > -1/e$, for $u > 0$, we obtain

$$L_1^{\pm} \geq e^{-\frac{z}{e \ln(n)}} \int_0^{\frac{t_1}{\|x - x_0\|^d}} e^{-u\left(1 + z \frac{\ln\left(D_\pm \|x - x_0\|^{-d}\right)}{\ln(n)}\right)} \, du, \quad (98)$$

$$\geq \frac{1}{1 + z} e^{-\frac{z}{e \ln(n)}} \left(1 - e^{-\frac{t_1}{\|x - x_0\|^d}}\right), \quad (99)$$

which shows that $L_1^{\pm}$ is bounded from below by a term for which the $\lim_Z$ is $\frac{1}{1+Z}$.

Anticipating that we will take the $\lim_Z$ and hence the limit $x_0 \to x$, we can freely assume that $\|x - x_0\| < 1$ and $K = \frac{t_1}{\|x - x_0\|^{d/2}} > 1$, so that we also have $K < \frac{t_1}{\|x - x_0\|^d}$. We then obtain

$$L_1^{\pm} \leq \int_0^K e^{-u\left(1 + z \frac{\ln\left(D_\pm \|x - x_0\|^{-d}/u\right)}{\ln(n)}\right)} \, du + \int_K^{+\infty} e^{-u} \, du, \quad (100)$$

$$\leq \int_0^1 e^{-u\left(1 + z \frac{\ln\left(D_\pm \|x - x_0\|^{-d}\right)}{\ln(n)}\right)} \, du + \int_1^K e^{-u\left(1 + z \frac{\ln\left(D_\pm \|x - x_0\|^{-d}/K\right)}{\ln(n)}\right)} \, du + e^{-K}, (101)$$

$$\leq \frac{1 - e^{-1 - z \frac{\ln\left(D_\pm \|x - x_0\|^{-d}\right)}{\ln(n)}}}{1 + z \frac{\ln\left(D_\pm \|x - x_0\|^{-d}\right)}{\ln(n)}} + \frac{e^{-1 - z \frac{\ln\left(D_\pm \|x - x_0\|^{-d}/K\right)}{\ln(n)}}}{1 + z \frac{\ln\left(D_\pm \|x - x_0\|^{-d}/K\right)}{\ln(n)}} + e^{-K}. \quad (102)$$

For $Z > 0$, $\lim_Z \frac{\ln\left(\|x - x_0\|^{-d}\right)}{\ln(n)} = \lim_Z \frac{\ln\left(\|x - x_0\|^{-d/2}\right)}{\ln(n)} = 1$, and the $\lim_Z$ of the upper bound in Eq. (102) is also $\frac{1}{1+Z}$. For $Z = 0$, we have $\lim_Z z \frac{\ln\left(\|x - x_0\|^{-d}\right)}{\ln(n)} = \lim_Z z \frac{\ln\left(\|x - x_0\|^{-d/2}\right)}{\ln(n)} = 0$, so that the $\lim_Z$ of the upper bound in Eq. (102) is 1. Finally, since $\lim_Z L_2 = 0$, we have shown

589 that for any real $Z \geq 0$, $\lim_Z L_1^{\pm} = \lim_Z L_0(x) = \frac{1}{1+Z}$, which proves Theorem 3.2. Note that the

590 two bounds obtained suggest that the relative error between $L_0(x)$ and $\frac{1}{1+Z}$ for finite large $n$ and

591 large $\|x - x_0\|^{-d}$ with $z(n, x_0)$ remaining close to $Z$ is of order $1/\ln(n)$, or equivalently, of order

592 $1/\ln(\|x - x_0\|)$.

593 *Numerical simulations for the Lagrange function at finite $n$*

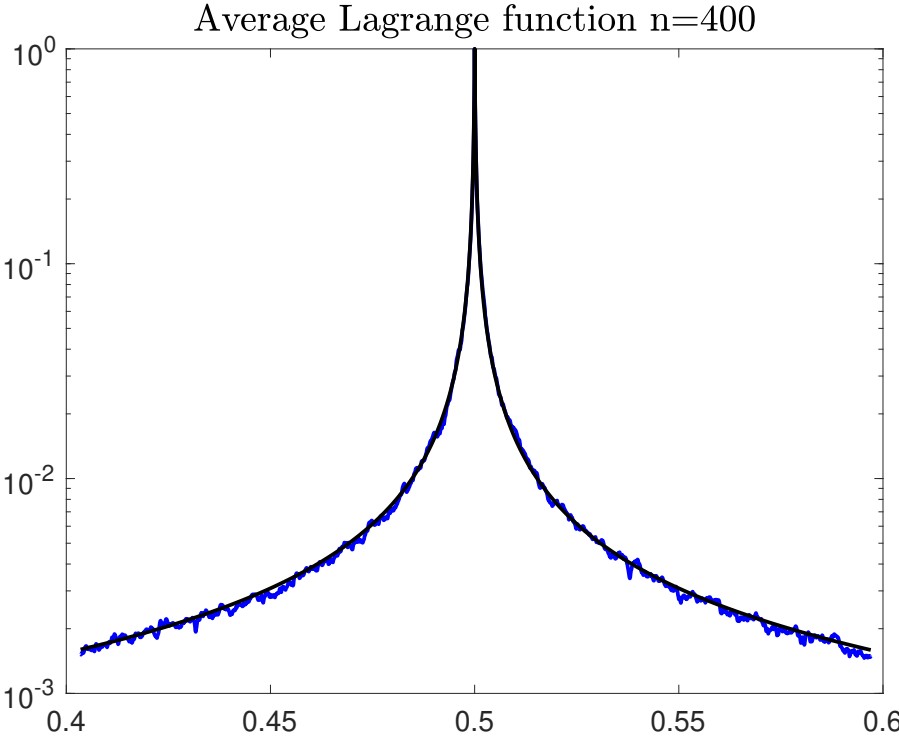

Figure 3: A numerical simulation is shown of the expected value of the Lagrange function of the Hilbert kernel regression estimator in one dimension for a uniform distribution like in Fig. 1. A total of $n = 400$ samples $x_i$ were chosen uniformly distributed in the interval $[0, 1]$ for 100 repeats and the Lagrange function evaluated at $x_0 = 0.5$ was averaged across these 100 repeats (blue curve). The black curve shows the asymptotic form $(1 + Z)^{-1}$ with $Z = 2|x - x_0|/W_n$. Since $n = 400$ is not too large, we used the implicit form for the scale $W_n$ given by $W_n \ln(1/W_n) = 1/n$ (see main text below Theorem 3.1) leading to $W_n^{-1} = 3232.39$ (compare with $400 \ln(400) = 2396.59$).

## A.4 The variance term

595 We define the variance term $\mathcal{V}(\text{x})$ as

$$\mathcal{V}(x) = \mathbb{E}\left[\sum_{i=0}^{n} w_i^2(x)[y_i - f(x_i)]^2\right] = \mathbb{E}_X\left[\sum_{i=0}^{n} w_i^2(x)\sigma^2(x_i)\right] = (n+1)\mathbb{E}\left[w_0^2(x)\sigma^2(x_0)\right]. \quad (103)$$

596 If we first assume that $\sigma^2(x)$ is bounded by $\sigma_0^2$, we can readily bound $\mathcal{V}(x)$ using Theorem 3.1 with

597 $\beta = 2$:

$$\mathcal{V}(x) \leq (n + 1)\sigma_0^2 \mathbb{E}\left[w_0^2(x)\right]. \quad (104)$$

598 Hence, for any $\varepsilon > 0$, there exists a constant $N_{x,\varepsilon}$, such that for $n \geq N_{x,\varepsilon}$, we obtain Theorem 3.3

$$\mathcal{V}(x) \leq (1 + \varepsilon)\frac{\sigma_0^2}{\ln(n)}. \quad (105)$$

599 However, one can obtain an exact asymptotic equivalent for $\mathcal{V}(x)$ by assuming that $\sigma^2$ is continuous

600 at $x$ (with $\sigma^2(x) > 0$), while relaxing the boundedness condition. Indeed, we now assume the growth

condition $C^\sigma_{\text{Growth}}$

$$\int \rho(y) \frac{\sigma^2(y)}{1 + \|y\|^{2d}}\, d^d y < \infty. \tag{106}$$

Note that this condition can be satisfied even in the case where the mean variance $\int \rho(y)\sigma^2(y)\, d^d y$ is infinite.

Proceeding along the very same line as the proof of Theorem 3.1 in section A.2, we can write

$$\mathbb{E}\left[ w_0^2(x)\sigma^2(x_0) \right] = \int_0^{+\infty} t\psi^n(x,t)\phi(x,t)\, dt, \tag{107}$$

with

$$\phi(x,t) := \int \rho(x+y)\sigma^2(x+y) \frac{e^{-\frac{t}{\|y\|^d}}}{\|y\|^{2d}}\, d^d y, \tag{108}$$

which as a similar form as Eq. (40), with $\beta = 2$. The condition of Eq. (106) ensures that the integral defining $\phi(x,t)$ converges for all $t > 0$.

The continuity of $\sigma^2$ at $x$ (and hence of $\rho\sigma^2$) and the fact the $\rho(x)\sigma^2(x) > 0$ implies the existence a small enough $\lambda > 0$ such that the ball $B(x,\lambda) \subset \Omega^\circ$ and $\|y\| \leq \lambda \implies |\rho(x+y)\sigma^2(x+y) - \rho(x)\sigma^2(x)| \leq \varepsilon\rho(x)\sigma^2(x)$, a property exploited for $\rho$ in the proof of Theorem 3.1 (see Eq. (52) and the paragraph above it), and which can now be used to efficiently bound $\phi(x,t)$. In addition, using the method of proof of Theorem 3.1 (see Eq. (64)) also requires that $\int_{\|y\|\geq\lambda} \rho(y)\frac{\sigma^2(y)}{\|y\|^{2d}}\, d^d y < \infty$, which is ensured by the condition $C^\sigma_{\text{Growth}}$ of Eq. (106). Apart from these details, one can proceed strictly along the proof and Theorem 3.1, leading to the proof of Theorem 3.4:

$$\mathcal{V}(x) \underset{n\to+\infty}{\sim} \frac{\sigma^2(x)}{\ln(n)}. \tag{109}$$

Note that if $\sigma^2(x) = 0$, one can straightforwardly show that for any $\varepsilon > 0$, and for $n$ large enough, one has

$$\mathcal{V}(x) \leq \frac{\varepsilon}{\ln(n)}, \tag{110}$$

while a more optimal estimate can be easily obtained if one specifies how $\sigma^2$ vanishes at $x$.

## A.5 The bias term

This section aims at proving Theorem 3.5, 3.6, and 3.7.

*Assumptions*

We first impose the following growth condition $C^f_{\text{Growth}}$ for $f(x) := \mathbb{E}[Y \mid X = x]$:

$$\int \rho(y) \frac{f^2(y)}{(1 + \|y\|^d)^2}\, d^d y < \infty, \tag{111}$$

which is obviously satisfied if $f$ is bounded. Since $\rho$ is assumed to have a second moment, condition $C^f_{\text{Growth}}$ is also satisfied for any function satisfying $|f(x)| \leq A_f\|y\|^{d+1}$ for all $y$, such that $\|y\| \geq R_f$, for some $R_f > 0$. Using the Cauchy-Schwartz inequality, we find that the condition $C^f_{\text{Growth}}$ also implies that

$$\int \rho(y) \frac{|f(y)|}{1 + \|y\|^d}\, d^d y < \infty. \tag{112}$$

In addition, for any $x \in \Omega^\circ$ (so that $\rho(x) > 0$), we assume that there exists a neighborhood of $x$ such that $f$ satisfies a local Hölder condition. In other words, there exist $\delta_x > 0$, $K_x > 0$, and $\alpha_x > 0$, such that the ball $B(0,\delta_x) \subset \Omega$, and

$$\|y\| \leq \delta_x \implies |f(x+y) - f(x)| \leq K_x\|y\|^{\alpha_x}, \tag{113}$$

which defines condition $C^f_{\text{Holder}}$.

*Definition of the bias term and preparatory results*

We define the bias term $\mathcal{B}(\mathrm{x})$ as

$$\mathcal{B}(x) = \mathbb{E}_X\left[\left(\sum_{i=0}^n w_i(x)[f(x_i) - f(x)]\right)^2\right] = (n+1)\mathcal{B}_1(x) + n(n+1)\mathcal{B}_2(x), \quad (114)$$

$$\mathcal{B}_1(x) = \frac{1}{n+1}\mathbb{E}_X\left[\sum_{i=0}^n w_i^2(x)[f(x_i) - f(x)]^2\right], \quad (115)$$

$$= \mathbb{E}_X\left[w_0^2(x)[f(x_0) - f(x)]^2\right], \quad (116)$$

$$\mathcal{B}_2(x) = \frac{1}{n(n+1)}\mathbb{E}_X\left[\sum_{0 \le i < j \le n} w_i(x)w_j(x)[f(x_i) - f(x)][f(x_i) - f(x)]\right], \quad (117)$$

$$= \mathbb{E}_X\left[w_0(x)w_1(x)[f(x_0) - f(x)][f(x_1) - f(x)]\right]. \quad (118)$$

Exploiting again Eq. (36) for $\beta = 2$ like we did in section A.2, we obtain

$$\mathcal{B}_1(x) = \int_0^{+\infty} t\,\psi^n(x,t)\chi_1(x,t)\,dt, \quad (119)$$

where $\psi(x,t)$ is again the function defined in Eq. (37), and where

$$\chi_1(x,t) := \int \rho(x+y)\mathrm{e}^{-\frac{t}{||y||^d}}\frac{(f(x+y) - f(x))^2}{||y||^{2d}}\,d^dy. \quad (120)$$

For any $t > 0$, and under condition $C^f_{\mathrm{Growth}}$, the integral defining $\chi_1(x,t)$ is well defined. Moreover, $\chi_1(x,t)$ is a strictly positive and strictly decreasing function of $t > 0$.

Now, defining $u_i = ||x - x_i||^{-d}$, $i = 0, ..., n$ and exploiting again Eq. (36) for $\beta = 2$, we can write

$$w_0(x)w_1(x) = u_0 u_1 \int_0^\infty t\,\mathrm{e}^{-(u_0+u_1)t - (\sum_{i=2}^n u_i)t}\,dt \quad (121)$$

Now taking the expectation value over the $n + 1$ independent variables, we obtain

$$\mathcal{B}_2(x) = \int_0^{+\infty} t\,\psi^{n-1}(x,t)\chi_2^2(x,t)\,dt, \quad (122)$$

where

$$\chi_2(x,t) := \int \rho(x+y)\mathrm{e}^{-\frac{t}{||y||^d}}\frac{f(x+y) - f(x)}{||y||^d}\,d^dy. \quad (123)$$

Again, for any $t > 0$, and under condition $C^f_{\mathrm{Growth}}$, the integral defining $\chi_2(x,t)$ is well defined. Note that, the integral defining $\chi_2(x,0)$ is well behaved at $y = 0$ under condition $C^f_{\mathrm{Holder}}$. Indeed, for $||y|| \le \delta_x$, we have $\frac{|f(x+y)-f(x)|}{||y||^d} \le K_x||y||^{-d+\alpha_x}$, which is integrable at $y = 0$ in dimension $d$. Note that, if $f(x + y) - f(x)$ were only decaying as $const./\ln(||y||)$, then $|\chi_2(x,t)| \sim const.\ln(|\ln(t)|) \to +\infty$, when $t \to 0$, and $\chi_2(x,0)$ would not exist (see the end of this section where we relax the local Hölder condition).

From now, we denote

$$\kappa(x) := \chi_2(x,0) = \int \rho(x+y)\frac{f(x+y) - f(x)}{||y||^d}\,d^dy. \quad (124)$$

Also note that $\kappa(x) = 0$ is possible even if $f$ is not constant. For instance, if $\Omega$ is a sphere centered at $x$ or $\Omega = \mathbb{R}^d$, if $\rho(x + y) = \hat{\rho}(||y||)$ is isotropic around $x$ and, if $f_x : y \mapsto f(x+y)$ is an odd function of $y$, then we indeed have $\kappa(x) = 0$ at the symmetry point $x$.

*Upper bound for $\mathcal{B}_1(x)$*

For $\varepsilon > 0$, we define $\lambda$ like in section A.2 and define $\eta = \min(\lambda, \delta_x)$, so that

$$\chi_1(x,t) \le (1 + \varepsilon)K_x\rho(x)\int_{||y|| \le \eta} \mathrm{e}^{-\frac{t}{||y||^d}}||y||^{2(\alpha_x - d)}\,d^dy + \Lambda_x, \quad (125)$$

$$\Lambda_x = \int_{||y|| \ge \eta} \rho(x+y)\frac{(f(x+y) - f(x))^2}{||y||^{2d}}\,d^dy, \quad (126)$$

where the constant $\Lambda_x < \infty$ under condition $C_{\text{Growth}}^f$. The integral in Eq. (125), can be written as

$$\int_{||y||\leq\eta} e^{-\frac{t}{||y||^d}} ||y||^{2(\alpha_x-d)} d^dy = S_d \int_0^\eta e^{-\frac{t}{r^d}} r^{2\alpha_x-d-1} dr, \tag{127}$$

$$= V_d t^{\frac{2\alpha_x}{d}-1} \int_{\frac{t}{\eta^d}}^{+\infty} u^{-\frac{2\alpha_x}{d}} e^{-u} du, \tag{128}$$

Hence, we find that $\chi_1(x,t)$ is bounded for $\alpha_x > d/2$. For $\alpha_x < d/2$, and for $t < t_1$ small enough, there exists a constant $M(2\alpha_x/d)$ so that $\chi_1(x,t) \leq M(2\alpha_x/d)t^{\frac{2\alpha_x}{d}-1}$. Finally, in the marginal case $\alpha_x = d/2$ and for $t < t_1$, we have $\chi_1(x,t) \leq M(1)\ln(1/t)$, for some constant $M(1)$.

Now, exploiting again the upper bound of $\psi(x,t)$ obtained in section A.2 and repeating the steps to bound the integrals involving $\psi^n(x,t)$, we find that, for $\alpha_x \neq d/2$, $\mathcal{B}_1(x)$ is bounded up to a multiplicative constant by

$$\int_0^{t_1} t^{\min\left(1,\frac{2\alpha_x}{d}\right)} e^{-nV_d\rho(x)t\ln\left(\frac{D_-}{t}\right)} dt \underset{n\to+\infty}{\sim} M'(2\alpha_x/d)\left(V_d\rho(x)n\ln(n)\right)^{-\min\left(2,\frac{2\alpha_x}{d}+1\right)}, \tag{129}$$

where $M'(2\alpha_x/d)$ is a constant depending only on $2\alpha_x/d$. In the marginal case, $\alpha_x = d/2$, $\mathcal{B}_1(x)$ is bounded up to a multiplicative constant by $n^{-2}\ln(n)$.

In summary, we find that

$$(n+1)\mathcal{B}_1(x) = \begin{cases} O\left(n^{-\frac{2\alpha_x}{d}}(\ln(n))^{-1-\frac{2\alpha_x}{d}}\right), & \text{for } d > 2\alpha_x \\ O\left(n^{-1}(\ln(n))^{-1}\right), & \text{for } d = 2\alpha_x \\ O\left(n^{-1}(\ln(n))^{-2}\right), & \text{for } d < 2\alpha_x \end{cases} \tag{130}$$

*Asymptotic equivalent for $\mathcal{B}_2(x)$*

Let us first assume that $\kappa(x) = \chi_2(x,0) \neq 0$. Then again, as shown in detail in section A.2, the integral defining $\mathcal{B}_2(x)$ is dominated by the small $t$ region, and will be asymptotically equivalent to

$$\mathcal{B}_2(x) = \int_0^{+\infty} t\,\psi^{n-1}(x,t)\chi_2^2(x,t)\,dt, \tag{131}$$

$$\underset{n\to+\infty}{\sim} \kappa^2(x)\int_0^{t_1} t\,e^{-nV_d\rho(x)t\ln\left(\frac{D_\pm}{t}\right)}\,dt, \tag{132}$$

$$\underset{n\to+\infty}{\sim} \left(\frac{\kappa(x)}{V_d\rho(x)n\ln(n)}\right)^2. \tag{133}$$

On the other hand, if $\kappa(x) = 0$, one can bound $\chi_2(x,t)$ (up to a multiplicative constant) for $t \leq t_1$ by the integral

$$\int_{||y||\leq\eta} \left(1 - e^{-\frac{t}{||y||^d}}\right) ||y||^{\alpha_x-d} d^dy = S_d \int_0^\eta \left(1 - e^{-\frac{t}{r^d}}\right) r^{\alpha_x-d} r^{d-1} dr, \tag{134}$$

$$= V_d t^{\frac{\alpha_x}{d}} \int_{\frac{t}{\eta^d}}^{+\infty} u^{-1-\frac{\alpha_x}{d}} \left(1 - e^{-u}\right) du. \tag{135}$$

Hence, for $\kappa(x) = 0$, we find that

$$n(n+1)\mathcal{B}_2(x) = O\left(n^{-\frac{2\alpha_x}{d}}(\ln(n))^{-2-\frac{2\alpha_x}{d}}\right). \tag{136}$$

*Asymptotic equivalent for the bias term $\mathcal{B}(x)$*

In the generic case $\kappa(x) \neq 0$, we find that $(n+1)\mathcal{B}_1(x)$ is always dominated by $n(n+1)\mathcal{B}_2(x)$, and we find the following asymptotic equivalent for $\mathcal{B}(x) = (n+1)\mathcal{B}_1(x) + n(n+1)\mathcal{B}_2(x)$:

$$\mathcal{B}(x) \underset{n\to+\infty}{\sim} \left(\frac{\kappa(x)}{V_d\rho(x)\ln(n)}\right)^2. \tag{137}$$

In the non-generic case $\kappa(x) = 0$, the bound for $(n+1)\mathcal{B}_1(x)$ in Eq. (130) is always more stringent than the bound for $n(n+1)\mathcal{B}_2(x)$ in Eq. (136), leading to

$$
\mathcal{B}(x) = \begin{cases}
O\left(n^{-\frac{2\alpha_x}{d}}(\ln(n))^{-1-\frac{2\alpha_x}{d}}\right), & \text{for } d > 2\alpha_x \\[2mm]
O\left(n^{-1}(\ln(n))^{-1}\right), & \text{for } d = 2\alpha_x \\[2mm]
O\left(n^{-1}(\ln(n))^{-2}\right), & \text{for } d < 2\alpha_x
\end{cases}
\tag{138}
$$

which prove the statements made in Theorem 3.5.

*Interpretation of the bias term $\mathcal{B}(x)$ for $\kappa(x) \neq 0$*

Here, we assume the generic case $\kappa(x) \neq 0$ and define $\bar{f}(x) = \mathbb{E}\left[\hat{f}(x)\right]$. We have

$$
\Delta(x) \quad := \quad \mathbb{E}\left[\sum_{i=0}^{n} w_i(x)(f(x_i) - f(x))\right] = \bar{f}(x) - f(x),
\tag{139}
$$

$$
\bar{f}(x) \quad = \quad \mathbb{E}\left[\sum_{i=0}^{n} w_i(x)f(x_i)\right] = (n+1)\mathbb{E}\left[w_0(x)f(x_0)\right].
\tag{140}
$$

By using another time Eq. (36), we find that

$$
\Delta(x) \quad = \quad (n+1)\int_0^{+\infty} \psi^n(x,t)\chi_2(x,t)\,dt,
\tag{141}
$$

$$
\underset{n\to+\infty}{\sim} \quad n\,\kappa(x)\int_0^{t_1} e^{-nV_d\rho(x)t\ln\left(\frac{D_\pm}{t}\right)}\,dt,
\tag{142}
$$

$$
\underset{n\to+\infty}{\sim} \quad \frac{\kappa(x)}{V_d\rho(x)\ln(n)}.
\tag{143}
$$

Comparing this result to the one of Eq. (137), we find that the bias $\mathcal{B}(x)$ is asymptotically dominated by the square of the difference $\Delta^2(x)$ between $\bar{f}(x) = \mathbb{E}\left[\hat{f}(x)\right]$ and $f(x)$:

$$
\mathcal{B}(x) \underset{n\to+\infty}{\sim} \left(\mathbb{E}\left[\hat{f}(x)\right] - f(x)\right)^2,
\tag{144}
$$

a statement made in Theorem 3.5.

*Relaxing the local Hölder condition*

We now only assume the condition $C_{\text{Cont.}}^{f}$ that $f$ is continuous at $x$ (but still assuming the growth conditions). We can now define $\delta_x$ such that the ball $B(x,\delta) \subset \Omega^\circ$ and $||y|| \leq \delta_x \implies |f(x+y) - f(x)| \leq \varepsilon$. Then, the proof proceeds as above but by replacing $K_x$ by $\varepsilon$, $\alpha_x$ by 0, and by updating the bounds for $\chi_1(x,t)$ (for which this replacement is safe) and $\chi_2(x,t)$ (for which it is not). We now find that for $0 < t \leq t_1$, with $t_1$ small enough

$$
0 \leq \chi_1(x,t) \quad \leq \quad \varepsilon(1+2\varepsilon)V_d\rho(x)t^{-1},
\tag{145}
$$

$$
|\chi_2(x,t)| \quad \leq \quad \varepsilon(1+2\varepsilon)V_d\rho(x)\ln\left(\frac{1}{t}\right).
\tag{146}
$$

As already mentioned below Eq. (123) where we provided an explicit counterexample, we see that relaxing the local Hölder condition does not guarantee anymore that $\lim_{t\to 0}|\chi_2(x,0)| < \infty$. With these new bounds, and carrying the rest of the calculation as in the previous sections, we ultimately find the following weaker result compared to Eq. (137) and Eq. (138):

$$
\mathcal{B}(x) = o\left(\frac{1}{\ln(n)}\right),
\tag{147}
$$

or equivalently, that for any $\varepsilon > 0$, there exists a constant $N_{x,\varepsilon}$ such that, for $n \geq N_{x,\varepsilon}$, we have

$$\mathcal{B}(x) \leq \frac{\varepsilon}{\ln(n)}. \tag{148}$$

*The bias term at a point where $\rho(x) = 0$*

This section aims at proving Theorem 3.7 expressing the lack of convergence of the estimator $\hat{f}(x)$ to $f(x)$, when $\rho(x) = 0$, and under mild conditions. Let us now consider a point $x \in \partial\Omega$ for which $\rho(x) = 0$, let us assume that there exists constants $\eta_x, \gamma_x > 0$, and $G_x > 0$, such that $\rho$ satisfies the local Hölder condition at $x$

$$||y|| \leq \eta_x \implies \rho(x + y) \leq G_x ||y||^{\gamma_x}. \tag{149}$$

We will also assume that the growth condition of Eq. (112) is satisfied. With these two conditions, $\kappa(x)$ defined in Eq. (124) exists. The vanishing of $\rho$ at $x$ strongly affects the behavior of $\psi(x,t)$ in the limit $t \to 0$, which is not singular anymore:

$$1 - \psi(x,t) \underset{t \to 0}{\sim} t \int \rho(y)||x - y||^{-d} d^d y, \tag{150}$$

where the convergence of the integral $\lambda(x) := \int \rho(y)||x - y||^{-d} d^d y$ is ensured by the local Hölder condition of $\rho$ at $x$.

Let us now evaluate $\bar{f}(x) = \lim_{n \to +\infty} \mathbb{E}[\hat{f}(x)]$, the expectation value of the estimator $\hat{f}(x)$ in the limit $n \to +\infty$, introduced in Eq. (140). First assuming, $\kappa(x) = \chi_2(x,0) \neq 0$, we obtain

$$\bar{f}(x) - f(x) = \lim_{n \to +\infty} (n+1) \int_0^{+\infty} \psi^n(x,t) \chi_2(x,t)\, dt, \tag{151}$$

$$= \lim_{n \to +\infty} n\, \chi_2(x,0) \int_0^{t_1} e^{n\, t\, \partial_t \psi(x,0)}\, dt, \tag{152}$$

$$= \frac{\kappa(x)}{\lambda(x)}, \tag{153}$$

which shows that the bias term does not vanish in the limit $n \to +\infty$. Eq. (153) can be straightforwardly shown to remain valid when $\kappa(x) = 0$. Indeed, for any $\varepsilon > 0$ chosen arbitrarily small, we can choose $t_1$ small enough such that $|\chi_2(x,t)| \leq \varepsilon$ for $0 \leq t \leq t_1$, which leads to $|\bar{f}(x) - f(x)| \leq \varepsilon/\lambda(x)$.

Note that relaxing the local Hölder condition for $\rho$ at $x$ and only assuming the continuity of $f$ at $x$ and $\kappa(x) \neq 0$ is not enough to guarantee that $\bar{f}(x) \neq f(x)$. For instance, if $\rho(x + y) \sim_{y \to 0} \rho_0/\ln(1/||y||)$, and there exists a local solid angle $\omega_x > 0$ at $x$, one can show that $1 - \psi(x,t) \sim_{t \to 0} \omega_x S_d \rho_0\, t \ln(\ln(1/t))$, and the bias would still vanish in the limit $n \to +\infty$, with $\hat{f}(x) - f(x) \sim_{n \to +\infty} \kappa(x)/[\omega_x S_d \rho_0 \ln(\ln(n))]$.

## A.6 Asymptotic equivalent for the regression risk

This sections aim at proving Theorem 3.8. Under conditions $C^\sigma_{\text{Growth}}$, $C^f_{\text{Growth}}$, and $C^f_{\text{Cont.}}$, the results of Eq. (109) and Eq. (147) show that for $\rho(x)\sigma^2(x) > 0$ and $\rho$ and $\sigma^2$ continuous at $x$, the bias term $\mathcal{B}(x)$ is always dominated by the variance term $\mathcal{V}(x)$ in the limit $n \to +\infty$. Thus, the excess regression risk satisfies

$$\mathbb{E}[(\hat{f}(x) - f(x))^2] \underset{n \to +\infty}{\sim} \frac{\sigma^2(x)}{\ln(n)}. \tag{154}$$

As a consequence, the Hilbert kernel estimate converges pointwise to the regression function in probability. Indeed, for $\delta > 0$, there exists a constant $N_{x,\delta}$, such that

$$\mathbb{E}[(\hat{f}(x) - f(x))^2] \leq (1 + \delta)\frac{\sigma^2(x)}{\ln(n)}, \tag{155}$$

for $n \geq N_{x,\delta}$. Moreover, for any $\varepsilon > 0$, since $\mathbb{E}[(\hat{f}(x) - f(x))^2] \geq \varepsilon^2\, \mathbb{P}[|\hat{f}(x) - f(x)| \geq \varepsilon]$, we deduce the following Chebyshev bound, valid for $n \geq N_{x,\delta}$

$$\mathbb{P}[|\hat{f}(x) - f(x)| \geq \varepsilon] \leq \frac{1 + \delta}{\varepsilon^2}\frac{\sigma^2(x)}{\ln(n)}. \tag{156}$$

 **A.7  Rates for the plugin classifier**

721 In the case of binary classification $Y \in \{0, 1\}$ and $f(x) = \mathbb{P}[Y = 1 \mid X = x]$. Let $F \colon \mathbb{R}^d \to \{0, 1\}$
722 denote the Bayes optimal classifier, defined by $F(x) := \theta(f(x) - 1/2)$ where $\theta(\cdot)$ is the Heaviside
723 theta function. This classifier minimizes the risk $\mathcal{R}_{0/1}(h) := \mathbb{E}[\mathbb{1}_{\{h(X) \neq Y\}}] = \mathbb{P}[h(X) \neq Y]$ under
724 zero-one loss. Given the regression estimator $\hat{f}$, we consider the plugin classifier $\hat{F}(x) = \theta(\hat{f}(x) - \frac{1}{2})$,
725 and we will exploit the fact that

$$0 \leq \mathbb{E}[\mathcal{R}_{0/1}(\hat{F}(x))] - \mathcal{R}_{0/1}(F(x)) \leq 2\,\mathbb{E}[|\hat{f}(x) - f(x)|] \leq 2\sqrt{\mathbb{E}[(\hat{f}(x) - f(x))^2]} \tag{157}$$

726 *Proof of Eq. (157)*

727 For the sake of completeness, let us briefly prove the result of Eq. (157). The rightmost inequality is
728 simply obtained from the Cauchy-Schwartz inequality and we hence focus on proving the first inequal-
729 ity. Obviously, Eq. (157) is satisfied for $f(x) = 1/2$, for which $\mathbb{E}[\mathcal{R}_{0/1}(\hat{F}(x))] = \mathcal{R}_{0/1}(F(x)) =$
730 $1/2$.

731 If $f(x) > 1/2$, we have $F(x) = 1$, $\mathcal{R}_{0/1}(F(x)) = 1 - f(x)$, and

$$\mathbb{E}[\mathcal{R}_{0/1}(\hat{F}(x))] = f(x)\mathbb{P}[\hat{f}(x) \leq 1/2] + (1 - f(x))\mathbb{P}[\hat{f}(x) \geq 1/2], \tag{158}$$

$$= \mathcal{R}_{0/1}(F(x)) + (2f(x) - 1)\mathbb{P}[\hat{f}(x) \leq 1/2], \tag{159}$$

732 which implies $\mathbb{E}[\mathcal{R}_{0/1}(\hat{F}(x))] \geq \mathcal{R}_{0/1}(F(x))$. Since $\mathbb{P}[\hat{f}(x) \leq 1/2] = \mathbb{E}[\theta(1/2 - \hat{f}(x))]$, and using
733 $\theta(1/2 - \hat{f}(x)) \leq \frac{|\hat{f}(x) - f(x)|}{f(x) - 1/2}$, valid for any $1/2 < f(x) \leq 1$, we readily obtain Eq. (157).

734 Similarly, in the case $f(x) < 1/2$, we have $F(x) = 0$, $\mathcal{R}_{0/1}(F(x)) = f(x)$, and

$$\mathbb{E}[\mathcal{R}_{0/1}(\hat{F}(x))] = \mathcal{R}_{0/1}(F(x)) + (1 - 2f(x))\mathbb{P}[\hat{f}(x) \geq 1/2]. \tag{160}$$

735 Since $\mathbb{P}[\hat{f}(x) \geq 1/2] = \mathbb{E}[\theta(\hat{f}(x) - 1/2)]$, and using $\theta(\hat{f}(x) - 1/2) \leq \frac{|\hat{f}(x) - f(x)|}{1/2 - f(x)}$, valid for any
736 $0 \leq f(x) < 1/2$, we again obtain Eq. (157) in this case.

737 In fact, for any $\alpha > 0$, the inequalities $\theta(1/2 - \hat{f}(x)) \leq \left(\frac{|\hat{f}(x) - f(x)|}{f(x) - 1/2}\right)^\alpha$ and $\theta(\hat{f}(x) - 1/2) \leq$
738 $\left(\frac{|\hat{f}(x) - f(x)|}{1/2 - f(x)}\right)^\alpha$ hold, respectively for $f(x) > 1/2$ and $f(x) < 1/2$. Combining this remark with the
739 use of the Hölder inequality leads to

$$\mathbb{E}[\mathcal{R}_{0/1}(\hat{F}(x))] - \mathcal{R}_{0/1}(F(x)) \leq 2|f(x) - 1/2|^{1-\alpha}\,\mathbb{E}\left[|\hat{f}(x) - f(x)|^\alpha\right], \tag{161}$$

$$\leq 2|f(x) - 1/2|^{1-\alpha}\,\mathbb{E}\left[|\hat{f}(x) - f(x)|^{\frac{\alpha}{\beta}}\right]^\beta, \tag{162}$$

740 for any $0 < \beta \leq 1$. In particular, for $0 < \alpha < 1$ and $\beta = \alpha/2$, we obtain

$$0 \leq \mathbb{E}[\mathcal{R}_{0/1}(\hat{F}(x))] - \mathcal{R}_{0/1}(F(x)) \leq 2|f(x) - 1/2|^{1-\alpha}\,\mathbb{E}\left[|\hat{f}(x) - f(x)|^2\right]^{\frac{\alpha}{2}}. \tag{163}$$

741 The interest of this last bound compared to the more classical bound of Eq. (157) is to show explicitly
742 the cancellation of the classification risk as $f(x) \to 1/2$, while still involving the regression risk
743 $\mathbb{E}\left[|\hat{f}(x) - f(x)|^2\right]$ (to the power $\alpha/2 < 1/2$).

744 *Bound for the classification risk*

745 Now exploiting the results of section A.6 for the regression risk, and the two inequalities Eq. (157)
746 and Eq. (163), we readily obtain Theorem 3.9.

747 **A.8  Extrapolation behavior outside the support of $\rho$**

748 This section aims at proving Theorem 3.10 characterizing the behavior of the regression estimator $\hat{f}$
749 outside the closed support $\hat{\Omega}$ of $\rho$ (extrapolation).

 *Extrapolation estimator in the limit $n \to \infty$*

751 We first assume the growth condition $\int \rho(y) \frac{|f(y)|}{1+\|y\|^d} \, d^d y < \infty$. For $x \in \mathbb{R}^d$ (i.e., not necessarily in
752 $\Omega$), we have quite generally

$$\mathbb{E}\left[\hat{f}(x)\right] = (n+1)\mathbb{E}\left[w_0(x)f(x)\right] = (n+1) \int_0^{+\infty} \psi^n(x,t)\chi(x,t)\, dt, \tag{164}$$

753 where $\psi(x,t)$ is again given by Eq. (37) and

$$\chi(x,t) := \int \rho(x+y)f(x+y)\frac{e^{-\frac{t}{\|y\|^d}}}{\|y\|^d}\, d^d y, \tag{165}$$

754 which is finite for any $t > 0$, thanks to the above growth condition for $f$.

755 Let us now assume that the point $x$ is not in the closed support $\bar{\Omega}$ of the distribution $\rho$ (which excludes
756 the case $\Omega = \mathbb{R}^d$ ). Since the integral in Eq. (164) is again dominated by its $t \to 0$ behavior, we have
757 to evaluate $\psi(x,t)$ and $\chi(x,t)$ in this limit, like in the different proofs above. In fact, when $x \notin \bar{\Omega}$,
758 the integral defining $\psi(x,t)$ and $\chi(x,t)$ are not singular anymore, and we obtain

$$1 - \psi(x,t) \underset{t\to 0}{\sim} t\int \rho(y)\|x-y\|^{-d}\, d^d y, \tag{166}$$

$$\chi(x,0) = \int \rho(y)f(y)\|x-y\|^{-d}\, d^d y. \tag{167}$$

759 Note that $\psi(x,t)$ has the very same linear behavior as in Eq. (150), when we assumed $x \in \partial\Omega$ with
760 $\rho(x) = 0$, and a local Hölder condition for $\rho$ at $x$.

761 Finally, by using the same method as in the previous sections to evaluate the integral of Eq. (164) in
762 the limit $n \to +\infty$, we obtain

$$\int_0^{+\infty} \psi^n(x,t)\chi(x,t)\, dt \underset{n\to+\infty}{\sim} \chi(x,0)\int_0^{t_1} e^{n\,t\,\partial_t\psi(x,0)}\, dt, \tag{168}$$

$$\underset{n\to+\infty}{\sim} \frac{1}{n}\frac{\chi(x,0)}{|\partial_t\psi(x,0)|}, \tag{169}$$

763 which leads to the first result of Theorem 3.10:

$$\hat{f}_\infty(x) := \lim_{n\to+\infty} \mathbb{E}\left[\hat{f}(x)\right] = \frac{\int \rho(y)f(y)\|x-y\|^{-d}\, d^d y}{\int \rho(y)\|x-y\|^{-d}\, d^d y}. \tag{170}$$

764 Note that since the function $(x,y) \longmapsto \|x-y\|^{-d}$ is continuous at all points $x \notin \bar{\Omega}$, $y \in \Omega$, and
765 thanks to the absolute convergence of the integrals defining $\hat{f}_\infty(x)$, standard methods show that $\hat{f}_\infty$
766 is continuous (in fact, infinitely differentiable) at all $x \notin \bar{\Omega}$.

767 *Extrapolation far from $\Omega$*

768 Let us now investigate the behavior of $\hat{f}_\infty(x)$ when the distance $L := d(x,\Omega) = \inf\{\|x-y\|, \ y \in$
769 $\Omega\} > 0$ between $x$ and $\Omega$ goes to infinity, which can only happen for certain $\Omega$, in particular, when $\Omega$
770 is bounded. We now assume the stronger condition, $\langle|f|\rangle := \int \rho(y)|f(y)|\, d^d y < \infty$, such that the $\rho$-
771 mean of $f$, $\langle f \rangle := \int \rho(y)f(y)\, d^d y$, is finite. We consider a point $y_0 \in \Omega$, so that $\|x-y_0\| \geq L > 0$,
772 and we will exploit the following inequality, valid for any $y \in \Omega$ satisfying $\|y-y_0\| \leq R$, with
773 $R > 0$:

$$0 \leq 1 - \frac{L^d}{\|x-y\|^d} \leq \frac{\|x-y\|^d - L^d}{L^d} \leq \frac{(L+R)^d - L^d}{L^d} \leq e^{\frac{dR}{L}} - 1. \tag{171}$$

774 Now, for a given $\varepsilon > 0$, there exist $R > 0$ large enough such that $\int_{\|y-y_0\|\geq R} \rho(y)\, d^d y \leq \varepsilon/2$ and
775 $\int_{\|y-y_0\|\geq R} \rho(y)|f(y)|\, d^d y \leq \varepsilon/2$. Then, for such a $R$, we consider $L$ large enough such that the

above bound satisfies $e^{\frac{dR}{L}} - 1 \le \varepsilon \min(1/\langle|f|\rangle, 1)/2$. We then obtain

$$\left| L^d \int \rho(y)f(y)\|x-y\|^{-d}\, d^d y - \langle f \rangle \right| \le \left( e^{\frac{dR}{L}} - 1 \right) \int_{\|y-y_0\| \le R} \rho(y)|f(y)|\, d^d y \quad (172)$$

$$+ \int_{\|y-y_0\| \ge R} \rho(y)|f(y)|\, d^d y, \quad (173)$$

$$\le \frac{\varepsilon}{2\langle|f|\rangle} \times \langle|f|\rangle + \frac{\varepsilon}{2} \le \varepsilon, \quad (174)$$

which shows that under the condition $\langle|f|\rangle < \infty$, we have

$$\lim_{d(x,\Omega) \to +\infty} d^d(x,\Omega) \int \rho(y)f(y)\|x-y\|^{-d}\, d^d y = \langle f \rangle. \quad (175)$$

Similarly, one can show that

$$\lim_{d(x,\Omega) \to +\infty} d^d(x,\Omega) \int \rho(y)\|x-y\|^{-d}\, d^d y = \int \rho(y)\, d^d y = 1. \quad (176)$$

Finally, we obtain the second result of Theorem 3.10,

$$\lim_{d(x,\Omega) \to +\infty} \hat{f}_\infty(x) = \langle f \rangle. \quad (177)$$

*Continuity of the extrapolation*

We now consider $x \notin \bar{\Omega}$ and $y_0 \in \partial\Omega$, but such that $\rho(y_0) > 0$ (i.e., $y_0 \in \partial\Omega \cap \Omega$), and we note $l := \|x - y_0\| > 0$. We assume the continuity at $y_0$ of $\rho$ and $f$ as seen as functions restricted to $\Omega$, i.e., $\lim_{y \in \Omega \to y_0} \rho(y) = \rho(y_0)$ and $\lim_{y \in \Omega \to y_0} f(y) = f(y_0)$. Hence, for any $0 < \varepsilon < 1$, there exists $\delta > 0$ small enough such that $y \in \Omega$ and $\|y - y_0\| \le \delta \implies |\rho(y_0) - \rho(y)| \le \varepsilon$ and $|\rho(y_0)f(y_0) - \rho(y)f(y)| \le \varepsilon$. Since we intend to take $l > 0$ arbitrary small, we can impose $l < \delta/2$.

We will also assume that $\partial\Omega$ is smooth enough near $y_0$, such that there exists a strictly positive local solid angle $\omega_0$ defined by

$$\omega_0 = \lim_{r \to 0} \frac{1}{V_d \rho(y_0) r^d} \int_{\|y-y_0\| \le r} \rho(y)\, d^d y = \lim_{r \to 0} \frac{1}{V_d r^d} \int_{y \in \Omega / \|y-y_0\| \le r} d^d y, \quad (178)$$

where the second inequality results from the continuity of $\rho$ at $y_0$ and the fact that $\rho(y_0) > 0$. If $y_0 \in \Omega^\circ$, we have $\omega_0 = 1$, while for $y_0 \in \partial\Omega$, we have generally $0 \le \omega_0 \le 1$. Although we will assume $\omega_0 > 0$ for our proof below, we note that $\omega_0 = 0$ or $\omega_0 = 1$ can happen for $y_0 \in \partial\Omega$. For instance, we can consider $\Omega_0, \Omega_1 \subset \mathbb{R}^2$ respectively defined by $\Omega_0 = \{(x_1, x_2) \in \mathbb{R}^2 / x_1 \ge 0, |x_2| \le x_1^2\}$ and $\Omega_1 = \{(x_1, x_2) \in \mathbb{R}^2 / x_1 \le 0\} \cup \{(x_1, x_2) \in \mathbb{R}^2 / x_1 \ge 0, |x_2| \ge x_1^2\}$. Then, it is clear that the local solid angle at the origin $O = (0,0)$ is respectively $\omega_0 = 0$ and $\omega_0 = 1$. Also note that if $x$ is on the surface of a sphere or on the interior of a face of a hypercube (and in general, when the boundary near $x$ is locally an hyperplane; the generic case), we have $\omega_x = \frac{1}{2}$. If $x$ is a corner of the hypercube, we have $\omega_x = \frac{1}{2^d}$.

Returning to our proof, and exploiting Eq. (178), we consider $\delta$ small enough such that for all $0 \le r \le \delta$, we have

$$\left| \int_{y \in \Omega / \|y-y_0\| \le r} d^d y - \omega_0 V_d\, r^d \right| \le \varepsilon\, \omega_0 V_d\, r^d. \quad (179)$$

We can now use these preliminaries to obtain

$$(\rho(y_0)f(y_0) - \varepsilon)J(x) - C \le \int \rho(y)f(y)\|x-y\|^{-d}\, d^d y \le (\rho(y_0)f(y_0) + \varepsilon)J(x) + C, \quad (180)$$

$$(\rho(y_0) - \varepsilon)J(x) - C' \le \int \rho(y)\|x-y\|^{-d}\, d^d y \le (\rho(y_0) + \varepsilon)J(x) + C', \quad (181)$$

with

$$J(x) \quad := \quad \int_{y \in \Omega \, / \, ||y-y_0|| \leq \delta} ||x-y||^{-d} \, d^d y, \tag{182}$$

$$C \quad = \quad \left(\frac{2}{\delta}\right)^2 \int_{||y-y_0|| \geq \delta} \rho(y)|f(y)| \, d^d y, \tag{183}$$

$$C' \quad = \quad \left(\frac{2}{\delta}\right)^2. \tag{184}$$

Let us now show that $\lim_{l \to 0} J(x) = +\infty$. We define $N := [\delta/l] \geq 2$, where $[\,.\,]$ is the integer part, and we have $N \geq 2$, since we have imposed $l < \delta/2$. For $n \in \mathbb{N} \geq 1$, we define,

$$I_n := \int_{y \in \Omega / ||y-y_0|| \leq \delta/n} d^d y, \tag{185}$$

and note that we have

$$I_n - I_{n+1} = \int_{\substack{y \in \Omega / ||y-y_0|| \leq \delta/n, \\ ||y-y_0|| \geq \delta/(n+1)}} d^d y, \tag{186}$$

$$\left| I_n - \omega_0 V_d \left(\frac{\delta}{n}\right)^d \right| \leq \varepsilon \, \omega_0 V_d \left(\frac{\delta}{n}\right)^d. \tag{187}$$

We can then write

$$J(x) \quad \geq \quad \sum_{n=1}^{N} \frac{1}{\left(l + \frac{\delta}{n}\right)^d} (I_n - I_{n+1}), \tag{188}$$

$$\geq \quad \sum_{n=1}^{N} \left( \frac{1}{\left(l + \frac{\delta}{n+1}\right)^d} - \frac{1}{\left(l + \frac{\delta}{n}\right)^d} \right) I_{n+1} + \frac{I_1}{(l+\delta)^d} - \frac{I_{N+1}}{\left(l + \frac{\delta}{N+1}\right)^d}. \tag{189}$$

We have

$$\frac{I_1}{(l+\delta)^d} - \frac{I_{N+1}}{\left(l + \frac{\delta}{N+1}\right)^d} \quad \geq \quad \omega_0 V_d \left( (1-\varepsilon) \frac{1}{\left(1 + \frac{l}{\delta}\right)^d} - (1+\varepsilon) \frac{1}{\left(1 + \frac{(N+1)l}{\delta}\right)^d} \right), \tag{190}$$

$$\geq \quad \omega_0 V_d \left( (1-\varepsilon) \frac{2^d}{3^d} - (1+\varepsilon) \right) =: C'', \tag{191}$$

which defines the constant $C''$. Now using Eq. (187), $l < \delta/2$, $N = [\delta/l]$, and the fact that $(1+u)^d - 1 \geq d\,u$, for any $u \geq 0$, we obtain

$$J(x) \quad \geq \quad (1-\varepsilon) \, \omega_0 V_d \sum_{n=1}^{N} \frac{1}{\left(1 + \frac{(n+1)l}{\delta}\right)^d} \left( \left(\frac{l + \frac{\delta}{n}}{l + \frac{\delta}{n+1}}\right)^d - 1 \right) + C'', \tag{192}$$

$$\geq \quad (1-\varepsilon) \, \omega_0 S_d \sum_{n=1}^{N} \frac{1}{\left(1 + \frac{(n+1)l}{\delta}\right)^{d+1}} \frac{1}{n} + C'', \tag{193}$$

$$\geq \quad \frac{(1-\varepsilon) \, \omega_0 \, S_d}{\left(1 + \frac{(N+1)l}{\delta}\right)^{d+1}} \ln(N-1) + C'', \tag{194}$$

$$\geq \quad (1-\varepsilon) \, \omega_0 \left(\frac{2}{5}\right)^{d+1} S_d \ln\left(\frac{\delta}{l} - 2\right) + C''. \tag{195}$$

808 We hence have shown that $\lim_{l\to 0} J(x) = +\infty$. Note that we can obtain an upper bound for $J(x)$
809 similar to Eq. (193) in a similar way as above, and with a bit more work, it is straightforward to show
810 that we have in fact $J(x) \sim_{l\to 0} \omega_0 S_d \ln\left(\frac{\delta}{l}\right)$, a result that we will not need here.

811 Now, using Eq. (180) and Eq. (181) and the fact that $\lim_{l\to 0} J(x) = +\infty$, we find that

$$\int \rho(y) f(y) \|x - y\|^{-d} d^d y \underset{l\to 0}{\sim} \rho(y_0) f(y_0) J(x), \tag{196}$$

$$\int \rho(y) \|x - y\|^{-d} d^d y \underset{l\to 0}{\sim} \rho(y_0) J(x), \tag{197}$$

812 for $f(y_0) \neq 0$ (remember that $\rho(y_0) > 0$), while for $f(y_0) = 0$, we obtain $\int \rho(y) f(y) \|x -$
813 $y\|^{-d} d^d y = o(J(x))$. Finally, we have shown that

$$\lim_{x\notin\bar\Omega, x\to y_0} \hat f_\infty(x) = f(y_0), \tag{198}$$

814 establishing the continuity of the extrapolation and the last part of Theorem 3.10.