# OpenReview forum: "Parameter-free Statistically Consistent Interpolation: Dimension-independent Convergence Rates for Hilbert kernel regression"
_NeurIPS.cc/2021/Conference — NeurIPS 2021 Submitted_

### Official Review · Reviewer_zJY4 · 2021-06-30

**Rating:** 5
**Confidence:** 3

**Summary:**

This paper provides an asymptotic analysis of the Hilbert kernel interpolating estimator, at a fixed covariate point $x$. First, it is noted that, since the estimator is a Nadaraya-Watson-type estimator, the bias and variance of the estimator depend primarily on the distribution of the weights $w_i(x)$. The paper thus provides a fairly precise asymptotic characterization of the moments of the weights. This characterization is then used to bound the variance (by $O(\sigma^2(x)/\log(n)$) and the bias (by $O(1/\log(n)$). This also leads to a comparable bound on classification risk for a plug-in classifier based on Hilbert kernel regression.

**Limitations And Societal Impact:**

The paper could benefit from additional discussion of its limitations (e.g., assumptions that could be weakened or results that could be strengthened), but this is not a serious weakness. I don't foresee any negative societal impacts of this work.

**Main Review:**

Overall, I feel that the paper contains interesting and significantly novel results, but I found it difficult to understand the motivation and intuition behind several parts of the paper (esp. Sections 3.1.2 and 3.4, but for other sections as well). I also suggest adding more exposition regarding the overall conclusions and consequences of the results derived in the paper, even if this requires removing some of the technical content.

**Main Comments:**

1. The length and detail of the abstract makes it hard to efficiently understand the purpose of the paper. I suggest cutting the first few sentences ("Previously.... generalize well"), since these are well-understood in the context of interpolating regression, and would be better placed in the intro to the paper. I also suggest replacing the exact mathematical bounds (e.g., "$2|f(x) − 1/2|^{1−\alpha} (1 + \epsilon)^\alpha \sigma^\alpha(x) (ln(n))^{-\alpha/2}$") with higher-level descriptions, especially since some of the notation (e.g., $\epsilon$) is not defined here.

2. I couldn't understand the notation "$Z$" in Theorem 3.2. As far as I could tell, $$Z := V_d \rho(x) \lim_{n \to \infty} \lim_{y \to 0} y^d n \log(n) = 0.$$ Thus, I couldn't understand what Theorem 3.2 was saying. More broadly, I don't know the motivation for characterizing the Lagrange function $L_0$.

3. More discussion is needed regarding the convergence rates derived in this paper. The derived rates are quite different from those usually seen in nonparametric regression. Specifically, they are polynomial in $1/\log(n)$, with exponents independent of the dimension $d$. Hence, the convergence rates are described as "dimension-independent", in contrast to more common rates for nonparametric regression that are polynomial in $1/n$, but with exponents decaying with $d$. Some specific questions:
a. On the practical side, a polynomial rate in $1/log(n)$ is too slow to be useful with realistic sample sizes. since the paper shows that Hilbert kernel regression converges at this rate regardless of the problem dimension, is the main conclusion of the paper that Hilbert kernel regression is impractical even in low dimensions?
a. On the theoretical side, it is interesting that a rate can be obtained independently of the problem dimension. Such a rate would have to be slower than any polynomial in $1/n$, given standard minimax lower bounds for nonparametric regression, but it is unclear whether the derived rates are optimal. Do there exist any other papers providing comparable dimension-independent upper or lower bounds?

4. I don't understand the motivation for Section 3.4. Why do we care about the behavior of the estimator for values $x$ that will never be observed?

**Minor Comments:**
1. On Line 186, what is "P(W)"? Based on Eq. (11), it looks like P the density function of W, but this isn't clear from the text.
2. Typo on Line 48: "A key observations" -> "A key observation"

**Time Spent Reviewing:**

4

---

> ### Author Response · Authors · 2021-08-08
> **The reviewer finds that the paper contains “interesting and significant novel results”. The reviewer’s primary concern is about exposition, which we will address as suggested by adding more motivation, intuition and discussion of the consequences of the results. We address all technical points raised by the reviewer. We note that 1/log(n) (which governs convergence for the Hilbert estimator) may in fact be smaller than 1/n^(1/d), for practically relevant n and large d.**
>
> We are grateful to the reviewer for noting that the paper contains "interesting and significant novel results". The reviewer found it difficult to understand the motivation and intuition behind several parts of the paper, and suggests "adding more exposition regarding the overall conclusions and consequences of the results derived in the paper, even if this requires removing some of the technical content.". We are happy to do so - this is the general feedback from all the reviewers, and we plan to revise the paper accordingly, as also noted in response to reviewers 1 (mgwj) and 2 (GNs9).
>
> We address the specific points raised by the reviewer in turn:
>
> 1. The reviewer suggests streamlining the abstract by cutting the first few lines of the abstract, and also replacing the mathematical statements of the bounds with higher level descriptions. We are happy to do this in the revision.
>
> 2. The reviewer did not understand the notation "lim$_Z$" in Theorem 3.2. The reviewer appears to have misread the definition of this scaling limit in the text as a sequential limit, whereas it was meant to be a simultaneous limit. We will clarify the definition of this limit in the revision: the idea is that $y\rightarrow 0$ and $n\rightarrow \infty$ simultaneously while the product $V_d \rho(x) y^d n\log(n)$ is tending to a fixed nonzero constant $Z$ (this is commonly called a "scaling limit"). This limit is discussed in more detail in the current proof appendix A.3 (line 582-586).
>
> 3. The reviewer suggests that more discussion is needed regarding the convergence rates derived in the paper. This is also a feedback from reviewers 1 and 2, and we will add such discussion, some of which have already been presented in the responses to reviewers 1 and 2.
>
> "On the practical side, a polynomial rate of $1/\log(n)$ is too slow to be useful with realistic sample sizes". Response: As discussed at length in responses to reviewers 1 and 2. In fact, $1/\log(n)$ can be numerically smaller than the corresponding k-NN/wiNN factors of $n^{-2a/(2a+d)}$ (which also includes constants growing with $d$) for practically relevant values of $n$ and $d$. For example, consider $n=10^6$, $d=100$ and $a=1$. Then, $1/\log(n)=0.07$, which is an order of magnitude smaller than $n^{-2a/(2a+d)}=0.76$. Moreover, the constant prefactors for the  $n^{-2a/(2a+d)}$ error estimates generally grow rapidly with dimension. Actually, the reverse conclusion can be drawn: for high dimensional data sets, the convergence factors for estimators such as k-NN may be much bigger than the $1/\log(n)$ factor for the Hilbert estimator, for practically relevant choices of $n$ and $d$.
>
> 4. "On the theoretical side, it is interesting that a rate can be obtained independently of the problem dimension ... it is unclear whether the derived rates are optimal. Do there exist any other papers providing comparable dimension-independent upper or lower bounds?"
>
> Please note that we study a specific estimator (the Hilbert estimator) and obtain $exact~asymptotics$ for the error rates. Hence, the resulting bounds cannot be improved for this estimator, and it is therefore optimal for the Hilbert estimator. However, it is not clear whether other true non-parametric estimators could lead to tighter rates.
> As for other works, to our knowledge, this is the first work deriving the full convergence rates of the Hilbert estimator, and we are not aware of any other work deriving comparable dimension-independent asymptotic results (or upper or lower bounds) for nonparametric estimators of regression functions defined on finite dimensional spaces. On the other hand, reviewer 2 (GNs9) has pointed us to similar $1/\log(n)$ bounds for estimators for infinite dimensional spaces. We will cite these papers in the revision, and this points to the scope for future studies connecting regression estimation for functions defined on finite and infinite dimensional spaces.
>
> "is the main conclusion of the paper that Hilbert kernel regression is impractical even in low dimensions?" Response: In fact, as discussed above in point 3, the reverse is true: the comparative performance of the Hilbert kernel estimator (compared to conventional nonparametric estimators) is better and better in higher dimensions, and it is possible that in practical cases the Hilbert kernel outperforms conventional estimators in high enough dimensions. This is clearly a topic that deserves empirical study.
>
> 5. "I don't understand the motivation for Section 3.4. Why do we care about the behavior of the estimator for values that will never be observed?" Response: While input values outside the function domain will not be observed for training data, due to distributional shifts out-of-domain inputs may be observed in practice. The extrapolation behavior of interpolating estimators outside the domain in which training data exists is in fact an important practical issue of considerable current interest, which motivates our study of the extrapolation behavior of the Hilbert estimator.
>
> Minor comments:
>
> M1. As the reviewer notes, $P(W)$ is indeed the probability density function of $W$, which we will more explicitly state.
>
> M2. We will correct the language typo on line 48.
>
> Limitations: "The paper could benefit from additional discussion of its limitations (e.g., assumptions that could be weakened or results that could be strengthened), but this is not a serious weakness.". Response: We will emphasize more clearly the fact that the main limitation of our work is the assumption of a continuous density $\rho$  and a continuous $f$, but which allows us to treat many cases often neglected in the literature, as for instance when $x$ belongs to the boundary of the support, with $\rho(x)>0$ (relevance of the notion of solid angle) or even $\rho(x)=0$. Our growth conditions are in general optimal, and allow us to treat the case of an unbounded support, and/or unbounded $f$, $\rho$, and $\sigma$, contrary to many works in this field. As for the Holder conditions (for $\rho$ and/or $f$) appearing in several theorems, they are sufficient but not strictly necessary, but we present in the proof section several counterexamples obtained by replacing the Holder condition by just a continuity condition, showing that the Holder condition is "hard to beat" while remaining a simple, explicit, and standard condition. In the revised manuscript, we will express more explicitly the limitations but also the generality and strengths of our work.

---

> > ### Comment · Reviewer_zJY4 · 2021-09-10
> > **Thanks to the authors for their detailed response**
> >
> > I want to clarify my point regarding the "slow" convergence rate: My complaint was not that the convergence rate itself is slow (i.e., it did not detract from my perception of the paper), but rather that the implications of this rate were not discussed enough in the paper.
> >
> > I don't think it's very useful to argue that $1/\log(n)$ can be numerically smaller than $n^{-2a/(2a + d)}$ in high dimensions, since all of these bounds have many unknown constant factors, and ultimately, the question of whether the estimator can be useful in practice is better answered through simulations or experiments with real data than by theoretical analysis (which may therefore be worth adding in a revision of the paper!).
> >
> > In my opinion, conventional nonparametric estimators converge too slowly to be useful in high dimensions, so the main reason to compare performance with these estimators in high dimensions would be to draw a negative conclusion (i.e., that the Hilbert kernel estimator also converges to slowly to be useful). This kind of negative result is also an important finding, and I would support accepting a revision of this paper in which this finding is more clearly discussed. Also, it would help to discuss some of the theoretical context of existing results. For example, as I noted in my original review, the usual nonparametric minimax lower bounds (of order $n^{-2a/(2a + d)}$) imply that (a) conventional estimators already converge at the optimal rate and (b) any dimension-independent rate will have to be slower than any polynomial in $1/n$. These suggest to me that the $1/\log(n)$ rate might be (nearly) minimax-optimal, and I think this, together with the rates for regression over infinite-dimensional spaces mentioned by Reviewer GNs9, is worth discussing in the paper.
> >
> > Overall, I think that, with more extensive discussion, the paper has potential to be quite strong. Unfortunately, these revisions would require another round of review, and I therefore am retaining my score of 5.

---

> > > ### Author Response · Authors · 2021-09-20
> > > **Our MSE bound does not contain an arbitrary contstant, and the 1/log(n) rate is unrelated to the estimators for the infinite dimensional case**
> > >
> > > The reviewer states, "I don't think it's very useful to argue that can be numerically smaller than in high dimensions, since all of these bounds have many unknown constant factors". Respectfully, the reviewer is incorrect regarding the bound derived in our paper for Hilbert kernel regression. We provide an asymptotic equivalent for the excess risk *without any unknown constants* ($\sigma^2/\log(n)$). In fact, this is one of the interesting points of our paper. Thus, for $n=10^6$ the mean square error is given by the local noise variance reduced by a factor of $0.07$, a significant reduction.
> > >
> > > As for the corresponding $O(n^{-2a/(2a+d)})$ bounds, there are indeed prefactors that depend on specifics of the function classes involved (these are not mysterious and have analytical forms for specific function classes for the regression function). However, the point is that for reasonable $n$, $a$ and large $d$, $n^{-2a/(2a+d)}$ is very close to 1, and the seemingly significant power law factor (not a polynomial factor as the reviewer suggests) is practically not much of a reduction at all. In our example ($n=10^6$, $a=1$ and $d=100$), $n^{-2a/(2a+d)}=0.8$, fully a factor of 10 (!) larger than $1/\log(n)\sim 0.07$. Moreover, this situation does not improve until $\log(n)\sim 10^2$, which is an impossible ask in practice. Thus, the prefactor to the "better" power law term will have to compensate for a factor of 10 to be better than the $0.07$ variance reduction guaranteed by the $1\log(n)$ term. At the least, it is not obvious that the Hilbert estimator is automatically so much worse than the conventional estimators for high dimensions, that our results should be dismissed out of hand due to "too slow convergence rates".
> > >
> > >  We would like to also respectfully suggest that not every paper needs to have numerical experiments, which have their own problems of generality, as they depend on the specifics of the data sets in question. We respectfully suggest that the theorems presented in the paper are sufficiently strong (*e.g.* the lack of arbitrary constants for the regression estimator risk estimate) to merit further discussion by other attendees of the conference. The reviewer does not raise any technical objections to our paper that would require detailed re-review. It should be possible to judge the content of our revision, as it has been specifically delineated in our responses. We would also like to suggest that the point of a conference is to facilitate discussions between the paper authors and other conference attendees, if all discussions must be comprehensively presented in a conference manuscript of limited length (thus reducing space for technical results) it is not clear to us what the role of a conference is.
> > >
> > >  Finally, we have already addressed the commentary by reviewer GNs9 and shown that the $O(\log(n)^{-\xi})$ rates for infinite dimensional kernel regression estimators are not related to our $\sim \log(n)^{-1}$ rates for finite $d$.

---

### Official Review · Reviewer_pxdi · 2021-07-12

**Rating:** 9
**Confidence:** 4

**Summary:**

This paper studies the asymptotic behavior of the Hilbert kernel-based interpolation scheme. By asymptotic analysis, limits of basis functions, bias and variance terms, and the mean of predicted functions are derived.

**Limitations And Societal Impact:**

No special limitations are observed.

Societal impact: better understanding of the Hilbert kernel-based interpolation algorithm.

**Main Review:**

This paper proposes to use the Hilbert kernel for interpolation, and eventually expect the obtained function to generalize. To this end, the underlying regression function is assumed Holder smooth with growth condition, and the distribution of noise is assumed to have bounded or slowly growing point-wise variance function. The proposed algorithm is parameter-free and does not need training, but it is expected to consume a lot of time for prediction.

The main technique adopted is asymptotic analysis. Some technical assumptions are made for the analysis. This paper definitely helps to better understand the Hilbert kernel interpolation.

**Time Spent Reviewing:**

4 hours

---

> ### Author Response · Authors · 2021-08-08
> **We thank the reviewer for a strongly positive assessment of the manuscript.**
>
> We thank the reviewer for a strongly positive assessment of the paper and a succinct statement of the results of the paper and their implications.
>
> As such, there are no specific items to address in the review, except perhaps for one point raised by the reviewer, regarding the resources necessary for prediction. The reviewer is correct that a direct computation of the estimator would require using every data point, thus the computational complexity would grow linearly with the training data size. However, we think that this can be significantly speeded up by using approaches such as the fast multipole method or pre-computing the estimator on a suitable grid. We will add a sentence to the paper along these lines.

---

### Official Review · Reviewer_GNs9 · 2021-07-16

**Rating:** 5
**Confidence:** 4

**Summary:**

The paper studies convergence rates for point-wise kernel-based regression in $R^d$ with the Hilbert kernel. This kernel is parameter-free and singular, yielding an interpolator of the data. Various results are given for this estimator, in particular, it is shown that under some continuity and growth conditions on the regression function $f(x) = E[y|X=x]$ and the density $\rho$, the excess error follows the asymptotic rate of $\Theta(\sigma(x)^2/ \text{log}(n))$, where $\sigma(x)^2$ is the variance of $y$ at $x$.

**Ethical Concerns:**

None.

**Limitations And Societal Impact:**

Yes.

**Main Review:**

The results seem thorough, rich in detail, interesting, and important, especially for the goal of better understanding the contemporary issue of interpolating vs. generalization (although here the number of samples $>>$ dimension). Unfortunately however, the paper is not well written and very hard to follow. It suffers from unclear and lacking discussions and it is written more like a technical report than a conference paper.

As I see it, significantly more emphasize should be put on the context of the results and their connection to other results in the literature. For example, a rate of $O(\sigma^2/\text{log}(n))$ for the excess error is extremely slow as compared to non-interpolating methods such as $k$-NN or standard kernel methods in finite-dimensional spaces. In fact, such rates are what standard methods achieve in various infinite-dimensional spaces where one gets exponential small-ball probability; see, for example, Section 6 in the book "Nonparametric functional data analysis" by Ferraty & Vieu (2010) and the discussion at the end of Section 1 in "Universal consistency and rates of convergence of multiclass prototype algorithms in metric spaces" by Weiss & Gyorfi (2020). In some sense, this might explain why the bound you give is dimension free. It is also not clear why in Theorem 3.2 you consider the specific limit $lim_Z$, which exactly corresponds to the mentioned small-ball probability regime.

In the abstract and line 89, it is not clear what $\epsilon$ is. Anyway, the abstract should be shortened significantly and should not simply repeat the main text.

You write that "Despite the superficial resemblance, the wiNN and Hilbert Kernel estimators have quite different convergence rates,..." but do not discuss it.

In the paper, the support is defined as the set of points $x$ with density $\rho(x)>0$. This makes sense only when $\rho$ is assumed continuous, since when $\rho$ is not continuous this set is not well defined as $\rho$ can be changed on a set of measure zero without any consequences. A standard definition of the support is the set of points $x$ such that any ball around $x$ with positive radius has non-zero measure. Since all your results assume continuous $\rho$ (if I'm not mistaken), the nonstandard definition makes sense, but the such continuity assumption is not mentioned when the support $\Omega$ is defined in Section 2. In addition, the definitions with the notation suggesting "interior" and "boundary" of $\Omega$ are not identical to the standard definition.

Not clear why the growth conditions are chosen as they are and how they relate to other standard tail conditions in the regression literature.

In line 176, I believe the expectation should be on $E_X$ and not $E_{X|x_i}$.

I couldn't understand how the equality in line 194 follows.

It is not clear what is the purpose and implications of the discussion beginning in line 203 on the a.s. convergence of the weights $w_0(x)$. It is also not clear at the end whether the failure of such convergence is conjectured or a solid fact.

In Theorem 3.5, it is not readily clear if "where we have $|\kappa(x)| < \infty$" is an additional assumption or follows from the previous assumptions.

In line 280, $C^\sigma$ should be $C^\rho$.

In Theorem 3.7 you do not assume that $f$ is continuous. So the discussion following the theorem is not clear: the failure of convergence of $\hat{f}(x)$ to $f(x)$ at $x$ such that $\rho(x)=0$ can hold trivially by changing the value of $f$ at that point, without affecting the Bayes error. In addition, I couldn't fully interpret Theorem 3.7 given my comments above on the support. Do you assume condition (1) of a solid angle at $x$ in Theorem 3.7?

Anyway, given the estimator statistical performance, one might conclude that using this estimator is a bad choice.

**Time Spent Reviewing:**

7

---

> ### Author Response · Authors · 2021-08-08
> **We appreciate that the reviewer finds the results “interesting and important”. The reviewer’s primary concern is about presentation, which we are happy to improve based on the reviewer’s input. We will provide more contextual discussion, particularly regarding the results for infinite dimensional spaces which the reviewer refers to. We address all technical points raised by the reviewer in this response and will accordingly modify the manuscript.**
>
> We thank the reviewer for a thorough and technically detailed review, which we believe will help us significantly improve the manuscript. We appreciate that the reviewer finds the results "rich in detail, interesting and important, especially for the contemporary issue if interpolating vs. generalization". The reviewer's primary concerns are about the presentation, which we believe we can improve.
>
> Each reviewer has provided feedback on how to improve the presentation, and we are happy and willing to do so taking into account all the feedbacks. We acknowledge that the paper had focussed on presenting the technical results rather than contextualizing discussion - an important reason being the space limitations of the main manuscript. However, given that the reviewers do not raise questions regarding the technical soundness of the results, we are happy to move some parts of the technical material to the appendix to make room for more discussion of the results and their context. We address the reviewer's points in turn:
>
> 1. "a rate of $O(\sigma^2/\log(n))$ for the excess error is extremely slow as compared to non-interpolating methods such as k-NN ..". Response: as discussed in more detail in the response to reviewer 1 (please see this response), for practically relevant large $n$ and $d$, one can in fact have $1/\log(n)$ to be much smaller than $n^{-2a/(2a+d)}$ (for $n=10^6$, $d=100$ and $a=1$ we get $1/\log(n)=0.07$, which is an order of magnitude smaller than $n^{-2a/(2a+d)}=0.76$). Note that for $n^{-1/d}$ to be smaller than $1/\log(n)$, one needs $n$ to be exponentially large in $d$, which is practically unattainable for even modestly large $d$. Thus, in practical cases with large $d$ the Hilbert estimator may perhaps outperform the conventional estimators! This issue certainly deserves a future empirical study.
>
> 2. "such rates are what standard methods achieve in various infinite dimensional spaces ..". Response: We thank the reviewer for pointing us to these interesting references, which we will cite. As our results are about regression functions defined on finite dimensional spaces, these references do not constitute prior derivations of our results. However, it is interesting to note that one could study a scaling regime in which both $n,d\rightarrow\infty$ while maintaining $n^{-1/d}\propto 1/\log(n)$ (this would require $d\sim \log(n)/ \log(\log(n))$). In such a scaling limit, one could perhaps connect the different behaviors together. We will provide a contextualizing discussion along these lines about the estimators for infinite dimensional spaces pointed to by the reviewer. This opens up a line of work, and we hope the reviewer will agree, makes our work therefore relevant for presentation to the community in this conference.
>
> 3. "it is not clear why in Theorem 3.2 you consider the specific lim$_Z$". Response: Our results indicate that this is the correct scaling limit to consider for studying the asymptotic behavior of the average Lagrange function of the estimator. Note that the Lagrange function has a central role in interpolation theory, and therefore its scaling limit is of interest to study.
>
> "In the abstract and line 89, it is not clear what $\epsilon$ is. Anyway, the abstract should be shortened significantly and should not simply repeat the main text." Response: As noted, we will streamline the abstract. $\epsilon$ is an arbitrary small number in the relevant proof, and we agree with the referee that it was used without introducing the full context.
>
> "You write that "Despite the superficial resemblance, the wiNN and Hilbert Kernel estimators have quite different convergence rates,..." but do not discuss it." Response: wiNN has the same convergence rate as k-NN ($\sim n^{-2a/(2a+d)}$ for regression), compared to $1/\log(n)$ for Hilbert. We will note this explicitly.
>
> 4. Definition of support, interior and boundary: In the "Assumptions" section, we will make clear that all our results are obtained for a continuous density $\rho$ $before$ stating our definitions for the support interior and boundary (currently these definitions are stated before the continuity condition for $\rho$).
>
> 5. "Not clear why the growth conditions are chosen as they are and how they relate to other standard tail conditions in the regression literature." Response: The growth conditions chosen are the ones that are necessary and sufficient for the convergence results stated, given the continuity and in some cases Holder assumptions on the underlying functions. The conditions are generally weaker than the standard conditions in the regression literature. For example, Devroye et al. in their original proof of the consistency of the Hilbert estimator assume boundedness of the regression function and variance, a condition we can relax. Or, consider Belkin, Rakhlin and Tsybakov (2019), Theorem 1, where the density $\rho$ is bounded away from zero from below, a considerably stronger condition than the one we assume for the density, which is even allowed to go to zero. We also refer the Referee to our answer to the last point of Referee 4 (zJY4).
>
> 6. "In line 176, I believe the expectation should be $E_X$ and not $E_{X|x_0}$"; "I couldn't understand how the equality in line 194 follows"; "In line 280, $C^{\sigma}$ should be $C^{\rho}$". Response: We thank the reviewer for catching these typos, which we will correct in the revised version. As for the equality of line 194, it follows from matching the result of Eq. (14) and (9). Note that an exponent $\beta$ was missing at the beginning of Eq. (14), where one should read $E[W^\beta]$ instead of $E[W]$.
>
> 7. "It is not clear what is the purpose and implications of the discussion beginning in line 203 on the a.s. convergence of the weights. It is also not clear at the end whether the failure of such convergence is conjectured or a solid fact."
> Response: Devroye et al., in their original work on the Hilbert estimator, proves using an example that the estimator does not exhibit a.s. convergence. However, this proof by counterexample does not provide any intuition about why the estimator lacks a.s. convergence. We provide such intuition by studying the behavior of the moments for arbitrary powers $\beta>-1$ and conjecturing a scaling form for the probability density function for the weights, which we verify numerically. We show that the conjectured scaling form would explain the lack of a.s. convergence of the weights; please see also the discussion in response to reviewer 1. We will make clearer that our derivation of the lack of a.s. convergence is heuristic and based on the conjecture that the distribution of weights follow a particular scaling form (see the discussion below Theorem 3.1), a conjecture supported by numerics and the fact that it allows to recover our exact results for the moments of the weights (see Eq. (13-14) and the discussion there).
>
> 8. "In Theorem 3.5...": we will make it clearer that the property $\kappa(x)<\infty$ results from the growth condition $C_{\rm Growth}^f$ and $C_{\rm Holder}^f$ mentioned at the beginning of Theorem 3.5.
>
> 9. "In Theorem 3.7 you do not assume that f is continuous .."
>  etc. We apologize, as there was indeed an omission in stating Theorem 3.7. We should in fact assume a local Holder condition for $f$ ($f$ is hence continuous) for $\kappa(x)$ to be well-defined (along with the growth condition stated in Theorem 3.7). See Theorem 3.5 where the conditions are correctly stated.
>  In addition, under the assumed local Holder condition for  $\rho$ at $x$, we do not have to assume the existence of a solid angle for $\rho$. In fact, at the very end of section A.5 (line 706), we show that relaxing the Holder condition for $\rho$ and now assuming the existence of a solid angle, we can find a counterexample to theorem 3.7 (i.e. $\hat f(x)=f(x)$ even for $\kappa(x)\ne 0$). This shows that the sufficient Holder condition for $\rho$ in Theorem 3.7, although not necessary, would be hard to replace by a weaker (but simple enough) condition for Theorem 3.7 to remain true (and for sure, just assuming the existence of a solid angle is not enough).
>
>  10. "Anyway, given the estimator statistical performance, one might conclude that using this estimator is a bad choice."
>  Response: In fact, the $1/\log(n)$ (independent of $d$) performance may not be so bad after all, for practical $n$ and $d$. Please see previous discussion, and response to reviewer 1. We will provide a discussion regarding this issue in the revised manuscript.

---

> > ### Author Response · Authors · 2021-08-23
> > **Relation of our result to Nadaraya Watson kernel regressors for infinite dimensional function spaces**
> >
> > Follow-up response to reviewer #2:
> >
> > We followed up with the references that the reviewer pointed us to (Ferraty and Vieu, 2010 and Weiss and Gyorfi, 2020), in the context of $\log(n)^{-\xi}$ convergence rates for infinite dimensional spaces. While these results are very interesting, they do not appear to be related to our results, despite the appearance of $\log(n)^{-1}$ in both cases.
> >
> > In brief, Ferraty and Vieu study Nadaraya-Watson regression estimators for infinite dimensional input spaces, using kernels that depend on a bandwidth parameter $h$ (note the difference from the Hilbert estimator where there is no such bandwidth parameter: in this sense, the Hilbert estimator is truly non-parameteric, but the estimators noted in Ferraty and Vieu are not). The difference of the estimated regression function and the true regression function (assumed to be Holder-$\beta$) has a bias-variance decomposition of the form $O(h^\beta)+O(\sqrt{\log(n)/(n \phi_\epsilon(h))}$ (Eq.6.65 in Ferraty and Vieu, 2010), where $\phi_\epsilon(h)$ is a small-ball probability. To go further, it is necessary to have some knowledge of $\phi_\epsilon(h))$. In Weiss and Gyorfi, 2020 (Eq.15) it is noted that for some classes of stochastic processes, $\phi_\epsilon(h))=O(\exp(-h^{-\tau}\log(h)^{-\tau'}))$. Performing a bias-variance tradeoff over the bandwidth parameter $h$ one would then obtain $h=O(\log(n)^{-1/\tau})$, leading to a convergence rate of $O(\log(n)^{-\beta/\tau})$.
> >
> > This leads to a convergence rate given by a power of the inverse logarithm, but this has nothing to do with our result. In our case, there is no bandwidth parameter and the variance term generically dominates over the bias term, so there is no bias-variance tradeoff. The variance term itself is similar to $\log(n)^{-1}$.
> >
> > We would respectfully suggest that the results we present are unrelated to the references pointed out by the referee (although we are happy to cite these interesting references, and briefly provide the contextual discussion above). We hope this further makes the case for our being able to present our results to other researchers in this conference.

---

> > > ### Comment · Reviewer_GNs9 · 2021-08-23
> > > **Thank you for the clarification.**
> > >
> > > .

---

### Official Review · Reviewer_mgwj · 2021-07-20

**Rating:** 5
**Confidence:** 4

**Summary:**

This article investigates the generalization guarantees of the Nadaraya-Watson estimator. The focus was put on the study of the finite sample risk of the estimator for two learning tasks: regression and classification.
The contributions are theoretical: the excess risk at a point $x$ (in the interior of the domain) converges to $0$ at the rate $\mathcal{O}(\sigma(x)^2/\log^{\alpha}(n))$, where $\sigma(x)$ is the noise variance at the point $x$, $\alpha$ is a positive real number that depends on the learning task, and n is the number of nodes used for the interpolation. Most importantly, the convergence rate is independent of the dimension $d$. In the process, many interesting theoretical results were proven: the study of the asymptotic equivalents of the moments of the weight functions, along with a scaling limit of the associated Lagrange functions.

**Limitations And Societal Impact:**

No.

**Main Review:**

Although the theoretical results proved in this work are very interesting for the kernel-based interpolation community, yet the exposition of the results is not typical, and I believe it is not suitable for an ML conference for the following reasons:
- there is no discussion of the theoretical results. The rates of convergence are “slow” compared to the typical Monte-Carlo rate. I believe that this interesting dependency of the risk on $n$ deserves a discussion.
- the technical results on the moments of the weight functions were presented before the main results (Theorem 3.8 and Theorem 3.9) that are relevant to ML community .
- in Theorem 3.1. the asymptotic of the moments $\mathbb{E} w_{0}(x)^\beta$ were calculated for $\beta \in ]-1,+\infty [$. Yet, it seems that only the case $\beta =2$ is relevant for the analysis of the regression and classification risk.




**Time Spent Reviewing:**

7

---

> ### Author Response · Authors · 2021-08-08
> **We thank the reviewer for noting that we present interesting theoretical results. We respectfully suggest that theoretical results regarding interpolating estimators are of current interest to the ML community. The reviewer's primary concern is about exposition, which we are happy to address as the reviewer suggests. We argue that the 1/log(n) rate is not too "slow" in practical terms since the value of 1/log(n) can numerically be much smaller than 1/n^(1/d) when d is large and for practical n.**
>
> We thank the reviewer for noting that many interesting theoretical results are derived in the paper. We would respectfully suggest that the results are indeed of interest to the ML community and not just the data interpolation community. The reason is that interpolation has been recognized as a valid methodology in ML and other theoretical works are being presented in the interpolation regime. There are not many results yet on interpolation techniques for noisy training data that exhibit statistical consistency (of which the Hilbert interpolation scheme studied in this manuscript is an example). Thus, new and precise theoretical results, which exhibit qualitatively different convergence behavior than estimators studied previously, should be of interest.
>
> As for the rate of convergence being "slow", while we recognize why the reviewer might say this, i.e., $\sim 1/\log(n)$ is asymptotically slower than the typical $\sim n^{-2a/(2a+d)}$ (e.g., for Nadaraya-Watson estimators with sufficiently regular kernels, or k-NN regression estimators, with $a$ being the Holder exponent of the class of functions being learned). Nevertheless, note that for values of $n$ and $d$ relevant to contemporary ML, in fact we can surprisingly have $1/\log(n) << n^{-2a/(2a+d)}$. For example, consider $n=10^6$, $d=100$ and $a=1$. Then, $1/\log(n)=0.07$, which is an order of magnitude smaller than $n^{-2a/(2a+d)}=0.76$. Moreover, the constant prefactors for the  $n^{-2a/(2a+d)}$ error estimates generally grow rapidly with dimension.
> This raises the intriguing possibility that for practical cases relevant to ML, the Hilbert estimator may indeed be competitive with other more well known estimators. We thus believe that the Hilbert estimator deserves to be more broadly known and would certainly benefit from empirical study. We believe that it would be helpful for ML practitioners to know about our results, and that this is an appropriate conference venue for us to present our results.
>
> In response to the specific points raised by the reviewer:
>
> 1. The reviewer calls for more discussion of the theoretical results, in particular of the non-typical $1/\log(n)$ convergence rates. We will be happy to add such discussion, including the discussion presented above. As also suggested by another reviewer, we can make room for such contextualizing discussions by moving more of our technical materials to the appendix.
>
> 2. The reviewer points out that the technical results of the moments of the weights, are presented before the main results that are relevant to the ML community (Theorems 3.8 and 3.9). The rationale for this organization is that theorems 3.8 and 3.9 are consequences of the results on weight moments. However, in light of the reviewer's concern about presentation, we are happy to move Theorems 3.8 and 3.9 upfront, together with a contextualizing discussion as presented above, and refer to the weight moment results, which will be moved to a subsequent section.
>
> 3. The reviewer questions why we study the moments $E[w_0(x)^\beta]$ for arbitrary $\beta>-1$, while the risk calculations only require $\beta=2$. The reason is that the knowledge of all moments provides some insight into the full distribution of the weights, for which we provide a numerically well-validated conjecture in the paper (see line 187 in the main text and Figure 2 in the appendix). This conjecture leads to results in agreement with our exact moment calculation. This distribution in turn provides insight into an interesting property of the Hilbert estimator, namely the likely lack of almost-sure convergence.
>
> This property was proven by Devroye et al. in their original work on the Hilbert kernel by demonstrating a counterexample, but no further insight was provided. Our results indicate that the weight-distribution is long-tailed, which can help explain why a.s. convergence does not occur. Intuitively, while the weights are generally $\sim 1/(n \log(n))$, once in a while a point occurs very close to a sample point, at which the weight is constrained to be $1$. This likely produces a long tail and prevents a.s. convergence. We will add to our discussion in section 3.1.1 to help the readers grasp this rationale for studying the moments for arbitrary $\beta>-1$.

---

### Decision · Program_Chairs · 2021-09-27

**Decision:**

Reject

**Comment:**

The topic of the submission is the study of nonparametric interpolation approaches in the context of supervised learning on R^d with real/binary labels. Particularly, the authors focus on the Hilbert kernel estimator (HKE, (2)-(3)), and study its L^2 risk: they show (Theorem 3.8) that the excess risk of the HKE estimator at point x is O(\sigma^2(x)/log(n)) where n denotes the sample size and \sigma^2(x) is the conditional variance function, and similar result (Theorem 3.9) holds for the classification risk of the plugin classifier.

Unfortunately, as the reviewers pointed out:
1) The submission is not written in a didactic way, it is quite hard to follow.
2) Discussion and numerical demonstration of the relevance on the established result are completely missing.

Significant revision is required.